# Amelioration of non-alcoholic fatty liver disease by targeting adhesion G protein-coupled receptor F1 (*Adgrf1*)

**Mengyao Wu[1], Tak-Ho Lo[2], Liping Li[3], Jia Sun[3], Chujun Deng[2], Ka-Ying Chan[2], Xiang Li[2], Steve Ting-Yuan Yeh[4], Jimmy Tsz Hang Lee[5,6], Pauline Po Yee Lui[7], Aimin Xu[5,6], Chi-Ming Wong[2,6,8]***

[1]Department of Chemistry and Chemical Engineering, Guangzhou University, Guangzhou, China; [2]Department of Health Technology and Informatics, Hong Kong Polytechnic University, Hong Kong, Hong Kong; [3]Zhujiang Hospital, Southern Medical University, China, China; [4]Ionis Pharmaceuticals, Carlsbad, United States; [5]Department of Medicine, University of Hong Kong, Hong Kong, Hong Kong; [6]State Key Laboratory of Pharmaceutical Biotechnology, University of Hong Kong, Hong Kong, China; [7]Department of Orthopaedics and Traumatology, Chinese University of Hong Kong, Hong Kong, Hong Kong; [8]Hong Kong Polytechnic University, Shenzhen Research Institute, Hong Kong, China

## Abstract

**Background:** Recent research has shown that the adhesion G protein-coupled receptor F1 (*Adgrf1*; also known as *GPR110; PGR19; KPG_012; hGPCR36*) is an oncogene. The evidence is mainly based on high expression of *Adgrf1* in numerous cancer types, and knockdown *Adgrf1* can reduce the cell migration, invasion, and proliferation. *Adgrf1* is, however, mostly expressed in the liver of healthy individuals. The function of *Adgrf1* in liver has not been revealed. Interestingly, expression level of hepatic *Adgrf1* is dramatically decreased in obese subjects. Here, the research examined whether *Adgrf1* has a role in liver metabolism.

**Methods:** We used recombinant adeno-associated virus-mediated gene delivery system, and anti-sense oligonucleotide was used to manipulate the hepatic *Adgrf1* expression level in diet-induced obese mice to investigate the role of *Adgrf1* in hepatic steatosis. The clinical relevance was examined using transcriptome profiling and archived biopsy specimens of liver tissues from non-alcoholic fatty liver disease (NAFLD) patients with different degree of fatty liver.

**Results:** The expression of *Adgrf1* in the liver was directly correlated to fat content in the livers of both obese mice and NAFLD patients. Stearoyl-coA desaturase 1 (*Scd1*), a crucial enzyme in hepatic de novo lipogenesis, was identified as a downstream target of *Adgrf1* by RNA-sequencing analysis. Treatment with the liver-specific *Scd1* inhibitor MK8245 and specific shRNAs against *Scd1* in primary hepatocytes improved the hepatic steatosis of *Adgrf1*-overexpressing mice and lipid profile of hepatocytes, respectively.

**Conclusions:** These results indicate *Adgrf1* regulates hepatic lipid metabolism through controlling the expression of *Scd1*. Downregulation of *Adgrf1* expression can potentially serve as a protective mechanism to stop the overaccumulation of fat in the liver in obese subjects. Overall, the above findings not only reveal a new mechanism regulating the progression of NAFLD, but also proposed a novel therapeutic approach to combat NAFLD by targeting *Adgrf1*.

**Funding:** This work was supported by the National Natural Science Foundation of China (81870586), Area of Excellence (AoE/M-707/18), and General Research Fund (15101520) to CMW, and the National Natural Science Foundation of China (82270941, 81974117) to SJ.

*For correspondence:
chi-ming.cm.wong@polyu.edu.hk

## Editor's evaluation

These valuable findings presented by Wu et al. advance our understanding in novel cell signaling regulators of hepatic metabolism. The evidence supporting these conclusions is solid, utilizing in vivo and in vitro gain- and loss-of-function studies. This work will be of interest to biologists working in the field of hepatic steatosis.

## Introduction

Liver is a vital organ as it is the site for undergoing a number of crucial physiological processes, including digestion, metabolism, immunity, and storage of nutrients (*Younossi et al., 2016*). Over-storage of lipid in the hepatocytes not caused by alcohol is known as non-alcoholic fatty liver disease (NAFLD), which is the most common liver pathological condition with a worldwide prevalence of 25% (*Cotter and Rinella, 2020*). The development of NAFLD is contributed by many factors such as lipid metabolism disorders, over- or mal-nutrition, inflammation, virus infection, or liver injuries (*Peng et al., 2020*). NAFLD usually does not entail any symptoms at early stages. However, if left untreated, NAFLD accounts for approximately 85% of all chronic non-communicable diseases (NCDs), such as type 2 diabetes mellitus (T2DM), cardiovascular disease (CVD), and chronic kidney disease (CKD) (*Allen et al., 2019*; *Zhou et al., 2019*). In addition, NAFLD may progress to non-alcoholic steato-hepatitis (NASH) with fibrosis, cirrhosis, or even hepatocellular carcinoma (HCC) (*Rinella and Sanyal, 2016*). Therefore, NAFLD imposes high economic and social burdens in terms of work productivity, health-related life quality, and use of healthcare resources (*Zhou et al., 2019*).

Improvement in managing NAFLD helps to resolve at least partially the progression of these diseases. Stopping the progression of NAFLD by lifestyle modifications such as increasing physical exercise activity and reduction of hypercaloric diet is only effective during the early stages before there is fibrosis. No medication is available to reverse the excessive fat storage in the liver once NASH developed. Therefore, it is urgently needed to unravel the mechanisms of NAFLD in order to accelerate the development, implementation, and explore new targets for the development of diagnostic tests and cost-effective therapies.

G protein-coupled receptors (GPCRs) are the largest and most diverse family of membrane receptor that play important roles in regulating most cellular and physiological processes (*Yang et al., 2021*). GPCRs are major targets for currently approved drugs (*Hauser et al., 2017*; *Sriram and Insel, 2018*). A few GPCRs have been shown to play key roles in NAFLD, and modulating their activities to ameliorate liver-related metabolic syndrome was proposed as NAFLD treatment (*Yang and Zhang, 2021*; *Kurtz et al., 2021*). However, currently proposed targets for GPCR-medicated NAFLD treatment are not exclusively expressed in hepatocytes, thus the potential side effects on other organs should be considered.

Human *Adgrf1* was identified by phylogenetic analysis based on highly conserved amino acid sequences of the GPCR transmembrane domains in 2002 (*Fredriksson et al., 2002*). Mouse ortholog of h*Adgrf1* was identified by the same research team 2 y later (*Bjarnadóttir et al., 2004*) and various splice variants were detected in deep sequencing experiments (*Bjarnadóttir et al., 2007*; *Lum et al., 2010*). So far, most *Adgrf1*-related studies focused on its tumorigenicity. In general, overexpression of *Adgrf1* was observed in various cancers, and it was required to promote cancer cell survival, proliferation, and migration (*Liu et al., 2018*; *Zhu et al., 2019b*; *Shi and Zhang, 2017*; *Bhat et al., 2018*; *Ma et al., 2017*; *Nam et al., 2022*). Therefore, it was suggested that targeting *Adgrf1* may represent a new therapeutic strategy for anticancer treatment. It was also reported that *Adgrf1* is required for proper fetal brain development and amelioration of neuroinflammation (*Lee et al., 2016a*). However, *Adgrf1* is predominantly expressed in health adult livers. The function of hepatic *Adgrf1* remains unexplored.

In this study, we provide the first evidence that *Adgrf1* induces the expression of *Scd1*, which contributes to NAFLD. By HFD-induced NAFLD mouse model (*Nakamura and Terauchi, 2013*), it is also shown that the repression of hepatic *Adgrf1* expression is a potential protective mechanism of preventing overaccumulation of lipid in liver. Importantly, the findings not only reveal a new mechanism regulation of the progression of NAFLD, but also proposed a novel therapeutic approach to combat NAFLD by targeting *Adgrf1*.

**eLife digest** Being overweight or obese increases the risk of developing numerous medical conditions including non-alcoholic fatty liver disease (NAFLD), where excess fat accumulates in the liver. NAFLD is a major global health issue affecting about 25% of the world's population and, if left untreated, can lead to liver inflammation as well as serious complications such as type 2 diabetes, heart disease, and liver cancer.

Currently, there are no medications which specifically treat NFALD. Instead, only medications which help to manage the associated health complications are available. Therefore, a better understanding of NFALD is required to help to develop new strategies for diagnosing and treating the progression of this disease.

A family of proteins known as GPCRs have crucial roles in regulating various bodily processes and are therefore commonly targeted for the treatment of disease. By identifying the GPCRs specifically involved in liver fat accumulation, new treatments for NFALD could be identified. Previous studies identified a GPCR known as Adgrf1 that is mainly found in liver cells, but its role remained unclear.

To investigate the function of Adgrf1 in the liver, Wu et al. studied obese mice and human patients with NAFLD. The experiments showed that elevated levels of Adgrf1 in human and mouse livers led to increased fat accumulation. On the other hand, livers with lower levels of Adgrf1 exhibited reduced fat levels. A technique called RNA sequencing revealed that Adgrf1 induces expression of enzymes involved in fat synthesis, including a key regulator called Scd1. Treating mice with high levels of liver fat with molecules that inhibit Scd1 decreased the symptoms of Adgrf1-mediated fatty liver disease.

These findings suggest therapies that decrease the levels of Adgrf1 may help to stop too much fat accumulating in the liver of human patients who are at risk of developing NAFLD. Further research is needed to confirm the effectiveness and safety of targeting Adgrf1 in humans and to develop suitable candidate drugs for the task.

## Methods

### Animals

All animal procedures were approved by the Animal Subjects Ethics Sub-Committee of the Hong Kong Polytechnic University and were conducted in accordance with the guidelines of the Centralized Animals Facilities (ethics number: 19-20/78-HTI-R-OTHERS). In general, 8-week-old C57BL/6J male mice were housed in pathogen-free conditions at controlled temperature with a 12 hr light–dark cycle and access to food and water ad libitum. The 8-week-old male mice were divided into two groups and fed with either standard chow diet (STC, 18.3% protein, 10.2% fat, 71.5% carbohydrates, Research Diet Inc, New Brunswick, NJ) or high-fat diet (HFD, 20% protein, 45% fat, 35% carbohydrates, Research Diets Inc) for 8 wk. The sample size was calculated based on previous findings, suggesting that group sizes of n = 7–8 would be sufficient (*Sellmann et al., 2017*).

The recombinant adeno-associated virus vector rAAV2/8 transduction was conducted as described previously (*Lee et al., 2016b*; *Cheng et al., 2022*). Briefly, mice were tail vein injected with $3 \times 10^{11}$ rAAV2/8 vector harboring either green fluorescent protein (*GFP*) or *Adgrf1*. For antisense oligonucleotide (ASO) delivery, validated ASOs against mouse *Adgrf1* were provided by Ionis Pharmaceuticals and injected subcutaneously once a week at 5 mg/kg body weight to the mice. For *Scd1* inhibitor delivery, MK-8245 (MedChemExpress, NJ) was gavage at 10 mg/kg once a week (*Iida et al., 2018*). All measurements were carried out in a randomized order.

### Primary hepatocyte isolation and adenovirus infection

Primary hepatocytes from different groups of mice were isolated using a two-step perfusion method as previously described (*Xu et al., 2022*; *Huang et al., 2017*). Briefly, type II collagenase was perfused to mice at a flow rate of 10 ml/min. Liver was collected and mesh in serum-free DMEM and hepatocytes were pelleted by centrifugation. For the supernatant containing non-parenchymal cells (NPCs), gradient solutions of Percoll were used for extraction. For adenovirus viral infection experiments, serum-starved cells were infected with adenoviruses carrying mouse *Adgrf1* cDNA to overexpress *Adgrf1*. Similar adenoviral vectors encoding the *GFP* gene were used as controls.

## Cell culture and luciferase reporter assay

HEK293T cells were purchased from the ATCC (CRL-11268) with the cell line authentication test conducted by the Hong Kong University Pathology Laboratory, showing that the percent match is higher than 80% threshold for authentication and no mycoplasma contamination. HEK293T cells were cultured in DMEM supplemented with 10% FBS, 100 U/ml penicillin, and 100 µg/ml streptomycin at 37°C and 5% $CO_2$. HEK293T cells were seeded in 6-well plates and were transfected with pGL3-*Scd1* promoter and adenoviral vector expressing either *Adgrf1* (ADV-*Adgrf1*) or *GFP* (ADV-*GFP*) by using the transfection reagent (#E4981, Promega, WI), following the manufacturer's instruction. DHEA (C3270, APExBIO, TX) was added into cells at the concentration of 100 µM and incubated at 37°C for 24 hr. For the luciferase reporter assay, pRL-TK (Renilla luciferase) reporter plasmid was used as a transfection control. The luciferase assays were performed by using the Dual-Luciferase Reporter Assay System (#E1960, Promega).

## Biochemical analysis

For glucose profile measurement, the blood glucose and insulin level were measured by collecting the blood samples from the tip of the tail using Accu-Chek glucometer (Roche Diagnostics, IN) as described (*Wong et al., 2014*). For the glucose tolerance test (GTT), insulin tolerance test (ITT), and pyruvate tolerance test (PTT), mice were fasted overnight prior to intraperitoneal injection of glucose (1 g/kg body weight [BW]; Sigma, St. Louis, MO), 0.75 U/kg BW insulin (Novolin R, Novo Nordisk, Bagsvaerd, Denmark), or 1 g/kg BW pyruvate (Sigma). Blood glucose levels were measured from the tip of tail vein at 15, 30, 60, 90, and 120 min after injection. For plasma and hepatic lipid level, serum levels of triglyceride (TG) and total cholesterol (CHO) were measured using commercial kit (BioSino, Biotechnology and Science Inc, China) according to the manufacturer's instructions. Hepatic lipids were extracted using Folch methodology, and liver extract was dissolved in ethanol for TG and CHO measurement. Both serum and hepatic levels of free fatty acid (FFA) were measured using commercial kit (Solarbio, China) according to the manufacturer's instructions. For liver function assay, the alanine aminotransferase (ALT) and aspartate aminotransferase (AST) levels were measured in serum using commercial kits (Stanbio) according to the manufacturer's instructions.

## Histopathological and Western blot analysis

Hematoxylin and eosin (H&E) and Oil Red O (ORO) staining was performed on paraffin-embedded and frozen liver sections, respectively. Detailed procedures of H&E and ORO were described previously (*Ye et al., 2016*; *Chen et al., 2017*). Representative histopathological images were acquired with a light microscope (Olympus, Tokyo, Japan). Western blot analysis was performed as previously described (*Lee et al., 2016b*; *Cheng et al., 2022*; *Xu et al., 2022*; *Huang et al., 2017*). Briefly, total protein was extracted from tissues and cultured cells with RIPA lysis buffer (65 mM Tris-HCl pH 7.5, 150 mM NaCl, 1 mM EDTA, 1% NP-40, 0.5% sodium deoxycholate, and 0.1% SDS). Protein samples were then separated by gel electrophoresis and then transferred to PVDF membranes (IPVH00010, Merck Millipore, CA). The expression of protein was detected by a ChemiDoc MP Imaging System (Bio-Rad, Hercules, CA). Primary antibodies used are shown in *Supplementary file 1*.

## Quantitative real-time PCR analysis

Total RNA was extracted with RNAiso Plus (#9109, TakaRa Bio Inc, Shiga, Japan) as previously described. RNA was reverse transcribed into cDNA with PrimeScript RT reagent Kit (#RR037, TakaRa Bio Inc). cDNA was then amplified with TB green Premix Ex Taq II (Til Rnase H Plus) (##RR820A, TakaRa Bio Inc). The real-time PCR was conducted with a LightCycler 96 RT-qPCR System (Roche, Basel, Switzerland). The relative quantity of the targeted RNA was calculated through normalization to the quantity of the corresponding GAPDH mRNA level. Detailed primer sequences are listed in *Supplementary file 2*.

## Microarray and RNA sequencing

The livers of mice fed with either STC (n = 6) or HFD (n = 6) for 8 wk were sent to Kompetenzzentrum Fluoreszente Bioanalytik (Germany) for gene expression analysis using Affymetrix Mouse Exon 1.0 ST Array. RNA was extracted from the livers of mice treated with rAAV-*Adgrf1* and ASO-*Adgrf1* using RNeasy Kits (QIAGEN, Hilden, Germany) according to the instructions. RNA concentration was

quantified using NanoDrop 2000 Spectrophotometer (Thermo Scientific, Waltham) and RNA quality was assessed using Agilent 2100 Bioanalyzer (Agilent, Santa Clara). 10 µg of total RNA from liver with RNA integrity number (RIN) > 7 was used in RNA-seq. RNA-seq was performed by BGI and analyzed by Dr. Tom system (BGI, Shenzhen, China). A heat map was created based on $\log_2$ transformed counts from different samples. To be included in the heat map, genes were required to have at least 1000 counts, totaled over all samples, where the standard deviation of the $\log_2$ had to exceed 2.

### Human samples

Liver biopsy specimens were collected from nine biopsy-proven NAFLD patients (*Xu et al., 2022*). Liver sections with H&E staining were subjected to histological evaluation of steatosis. Simple steatosis was defined by the presence of macrovascular steatosis affecting at least 5% of hepatocytes without inflammatory foci and evidence of hepatocellular injury in the form of hepatocyte ballooning (*Brunt et al., 1999*). Individuals with a heavy alcohol-drinking history (≥40 g/day for up to 2 wk), drug-induced liver disease, and hepatitis virus infection were excluded from the study. Clinical parameters of individuals are summarized in *Supplementary file 1*. The human study was approved by the Zhujiang Hospital, Southern Medical University, Guangzhou, China (number: 2019-KY-097-01). Written informed consent was obtained from participants prior to their inclusion in the study.

### Lipogenesis assay

Lipogenesis assay was performed as previously described (*Akie and Cooper, 2015*). Primary hepatocytes were seeded into 6-well plates and the cells washed with warm PBS twice one night prior to the assay. The hepatocytes were changed to serum starvation medium with 100 nM insulin and incubated overnight at 37°C. The lipogenesis medium was made up of 100 nM insulin, 10 µM cold acetate, and 0.5 µCi $^3$H-acetate and added and incubated for 2 hr. Cells were then washed with PBS twice and 0.1 N HCl was used to lyse cells. Lipids were extracted by addition of 500 µl of 2:1 chloroform-methanol (v/v). Total lipid content was then calculated by measuring $^3$H activity.

### Statistical analysis

All experiments were performed at least twice, and each experimental group included n ≥ 7 mice. Representative data are presented as mean ± standard error of mean (SEM). All statistical analyses were performed with the GraphPad Prism software (version 9.0, GraphPad, CA). Statistical differences among the two groups was performed using the unpaired Student's *t*-test or Mann–Whitney test for the comparison of variables with or without normal distribution, respectively. Correlation between two groups was assessed by non-parametric Spearman's test. For multiple comparisons between three or more groups, one-way ANOVA with Bonferroni correction was conducted. In all statistical comparisons, p-values<0.05 were accepted as significant.

## Results

### *Adgrf1* mainly expresses in liver of mice, and its expression is downregulated after HFD treatment

Firstly, we used microarray analysis to examine the change in expression levels of hepatic GPCRs in mice after HFD treatment (*Supplementary file 3*). In the screening results, we found that *Adgrf1* is mainly expressed in liver and its expression is dramatically decreased in the HFD-fed mice compared to their STC-fed littermates. Remarkably, in agreement with previous studies (*Ma et al., 2017*; *Prömel et al., 2012*), *Adgrf1* is mainly expressed in the liver of adult mice (*Figure 1A and B*). The *Adgrf1* protein expression in those tissues were checked by western blot analysis. GRP110 proteins were mainly detected in liver and kidney samples (*Figure 1B*). Next, we used cell fractionation to identify the *Adgrf1*-expressing cells in liver (*Figure 1C*). The *Cd11b* mRNAs were used as markers for non-parenchymal cells (NPCs), and albumin mRNA for hepatocytes. Our cell fractionation clearly demonstrated that *Adgrf1* mRNA is mainly expressed in hepatocytes (*Figure 1C*). This finding was further supported by western blot analysis (*Figure 1D*). Remarkably, in agreement with the microarray data, during HFD treatment for 8 wk, the expression level of hepatic *Adgrf1* gradually declined and to almost undetectable level at week 8 as examined by RT-qPCR analysis (*Figure 1E*). In contrast, the mRNA levels of the NAFLD-related marker *Fgf21* were highly induced in the livers of HFD-fed mice at

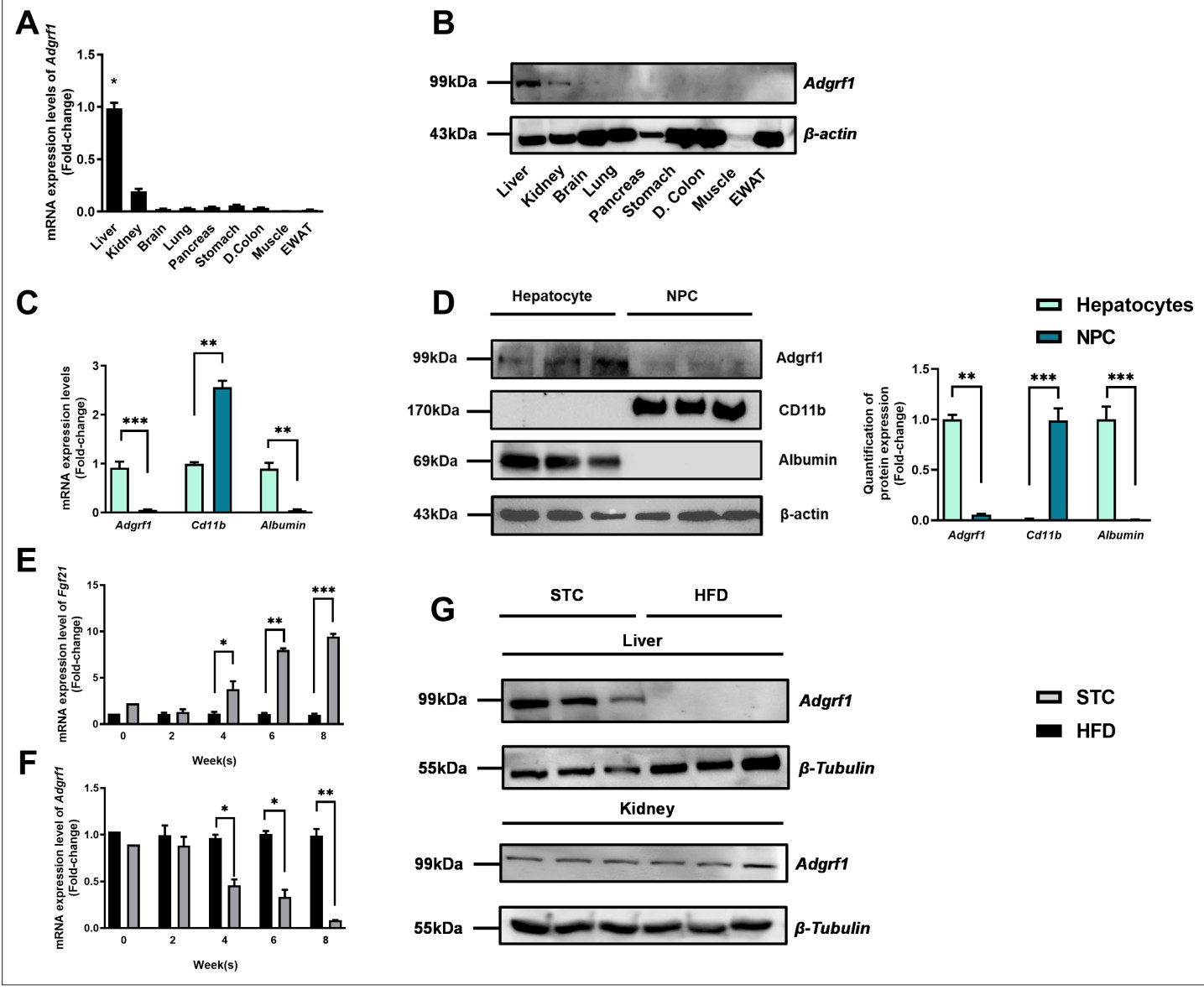

**Figure 1.** *Adgrf1* is mainly expressed in the liver and its expression is downregulated after HFD treatment. Eight-week-old male C57BL/6J mice were fed with either STC or HFD for 8 wk. (**A**) mRNA expression levels of *Adgrf1* in different organs as determined by RT-qPCR analysis (n = 5). (**B**) Representative immunoblotting analyses of Adgrf1 expression in different tissues of C57BL/J mice after STC for 8 wk (n = 3). (**C**) mRNA expression levels of *Adgrf1*, *Cd11b,* and albumin in factions of hepatocyte or NPCs isolated from STC-fed mice livers as determined by RT-qPCR. (**D**) Left panel: representative immunoblotting analyses of Adgrf1, Cd11b, and albumin in fractions of hepatocytes or NPC isolated from mice livers fed with STC, each lane is a sample from different individual; right panel: quantification of protein expression levels of Adgrf1, Cd11b, and albumin. Protein expression levels were normalized to the expression of *β-actin*. The fraction of hepatocytes was set as 1 for fold-change calculation. (**E**) mRNA expression levels of FGF21 in mice liver fed with 0, 2, 4, 6, and 8 wk of HFD as determined by RT-qPCR. (**F**) mRNA expression levels of Adgrf1 in mice liver fed with 0, 2, 4, 6, and 8 wk of HFD as determined by RT-qPCR. (**G**) Left panel: representative immunoblotting analyses of Adgrf1 in mice fed with either STC or HFD for 8 wk; right panel: quantification of protein expression levels of Adgrf1. Protein expression levels were normalized to the expression of β-tubulin. The sample from STC mice was set as 1 for fold-change calculation. Each lane is a sample from different individual. *Adgrf1*, G-protein-coupled receptor 110; STC, standard chow diet; HFD, high-fat diet; NPC, non-parenchymal cell. Data represented as mean ± SEM; repeated with three independent experiments; p-value analyzed by two-tailed Student's *t*-test. *$p<0.05$, **$p<0.01$, ***$p<0.001$.

The online version of this article includes the following source data for figure 1:

**Source data 1.** *Adgrf1* is mainly expressed in the liver and its expression is downregulated after HFD treatment.

week 8 (*Figure 1F*; *Rusli et al., 2016*). Western blot analysis was performed to confirm the declined expression of *Adgrf1* is also observed in protein level in the livers of HFD-fed mice (*Figure 1G*, upper panel). Interestingly, HFD treatment did not affect the renal GRP110 expression (*Figure 1G*, lower panel). Collectively, the hepatic, but not renal, *Adgrf1* level is tightly regulated by nutritional status.

## Overexpression of *Adgrf11* in hepatocytes accelerates metabolic dysregulation caused by HFD

Based on the dramatic difference in expression levels of hepatic *ADGRF1* before and after HFD treatment, we hypothesized that downregulation of *Adgrf1* in HFD-fed mice may be involved in the pathogenesis of fatty liver. To evaluate the impacts of high hepatic *Adgrf1* level on liver metabolism, *Adgrf1* was overexpressed in the hepatocytes of HFD-fed mice by liver-directed rAAV/thyroxine-binding globulin (TBG)-mediated gene expression system (*Figure 2A* and *Figure 2—figure supplement 1A*). The overexpression of *Adgrf1* in the livers of the mice was validated by RT-qPCR (*Figure 2B* and *Figure 2—figure supplement 1B*) and western blot analysis (*Figure 2—figure supplement 1C*). We also confirmed that rAAV-mediated *Adgrf1* overexpression was solely in hepatocytes, but not in NPC, by western blot analysis after cell fractionation (*Figure 2C*). Renal and adipose *Adgrf1* expression level was not affected by liver-directed rAAV/TBG-mediated gene expression (*Figure 2—figure supplements 1C and 2A*).

Overexpressing *Adgrf1* in the liver of STC-fed mice did not affect body weight (*Figure 2—figure supplement 1D*), fasting glucose level (*Figure 2—figure supplement 1E*), fasting insulin level (*Figure 2—figure supplement 1F*), and homeostatic model assessment for insulin resistance (HOMA-IR; *Figure 2—figure supplement 1G*). There was only a slight increase at several time points in the glucose excursion curve in response to the GTT (*Figure 2—figure supplement 1H*) and hepatic glucose production induced by sodium pyruvate in PTT (*Figure 2—figure supplement 1I*). No change in insulin sensitivity was observed by insulin tolerance test between STC-fed rAAV-*GFP* and rAAV-*Adgrf1* mice (*Figure 2—figure supplement 1J*).

However, under HFD treatment, rAAV-*Adgrf1* mice gained more body weight (*Figure 2D*) and body fat mass (*Figure 2E–F*) than their rAAV-*GFP* controls. The HFD-fed rAAV-*Adgrf1* mice also had higher fasting glucose level (*Figure 2G*), fasting insulin level (*Figure 2H*), and HOMA-IR (*Figure 2I*). Worsened glucose tolerance was observed in HFD-fed rAAV-*Adgrf1* mice (*Figure 2J*). Overexpression of *Adgrf1* in livers significantly increased hepatic glucose production induced in PTT (*Figure 2K*). ITT showed that the glucose levels in HFD-fed rAAV-*Adgrf1* mice remained insensitive at 30–60 min after injection of insulin compared to their control HFD-fed rAAV-*GFP* littermates (*Figure 2L*). In addition, we placed the rAAV-Adgrf1 mice into metabolic cages to explore their locomotor activities (*Figure 2—figure supplement 2B and C*), energy expenditure (*Figure 2—figure supplement 2D*), food intake (*Figure 2—figure supplement 2E*), water intake (*Figure 2—figure supplement 2F*), and respiratory exchange ratio (*Figure 2—figure supplement 2G*). In brief, there were no difference in these metabolic parameters between these HFD-fed rAAV-Adgrf1 and rAAV-GFP littermates (*Figure 2—figure supplement 2B–G*).

In summary, we observed a mild impairment in glucose homeostasis associated with overexpressing *Adgrf1* in the livers of STC-fed mice and more dramatic impairment was observed when the rAAV-*Adgrf1* mice was challenged with HFD compared to their rAAV-*GFP* controls.

## Suppressing *Adgrf1* improves glucose homeostasis in HFD-fed rAAV-*Adgrf1* mice

To confirm that the observations above were due to the rAAV-mediated overexpression of hepatic *Adgrf1* in HFD-fed mice, we used two N-acetylgalactaosamine (GalNAc) conjugated antisense oligonucleotides (ASO-*Adgrf1*s) that bind to different regions of *Adgrf1* mRNAs were used to knockdown the hepatic *Adgrf1* expression in mice (*Figure 3A* and *Figure 3—figure supplement 1A*). To avoid the observation is due to off-target effects, two different sequences of ASO were used. Chronic treatment of either ASO-*Adgrf1*s only lowered the hepatic, but not renal, *Adgrf1* mRNA (*Figure 3B* and *Figure 3—figure supplement 1B*) and protein (*Figure 3—figure supplement 1C*) levels. It is due to the fact that liver hepatocytes abundantly and specifically express the asialoglycoprotein receptor that binds and uptakes circulating glycosylated oligonucleotides via receptor-mediated endocytosis (*Cui et al., 2021*). Knockdown hepatic *Adgrf1* by ASO-*Adgrf1*s in STC-fed mice did not affect body weight

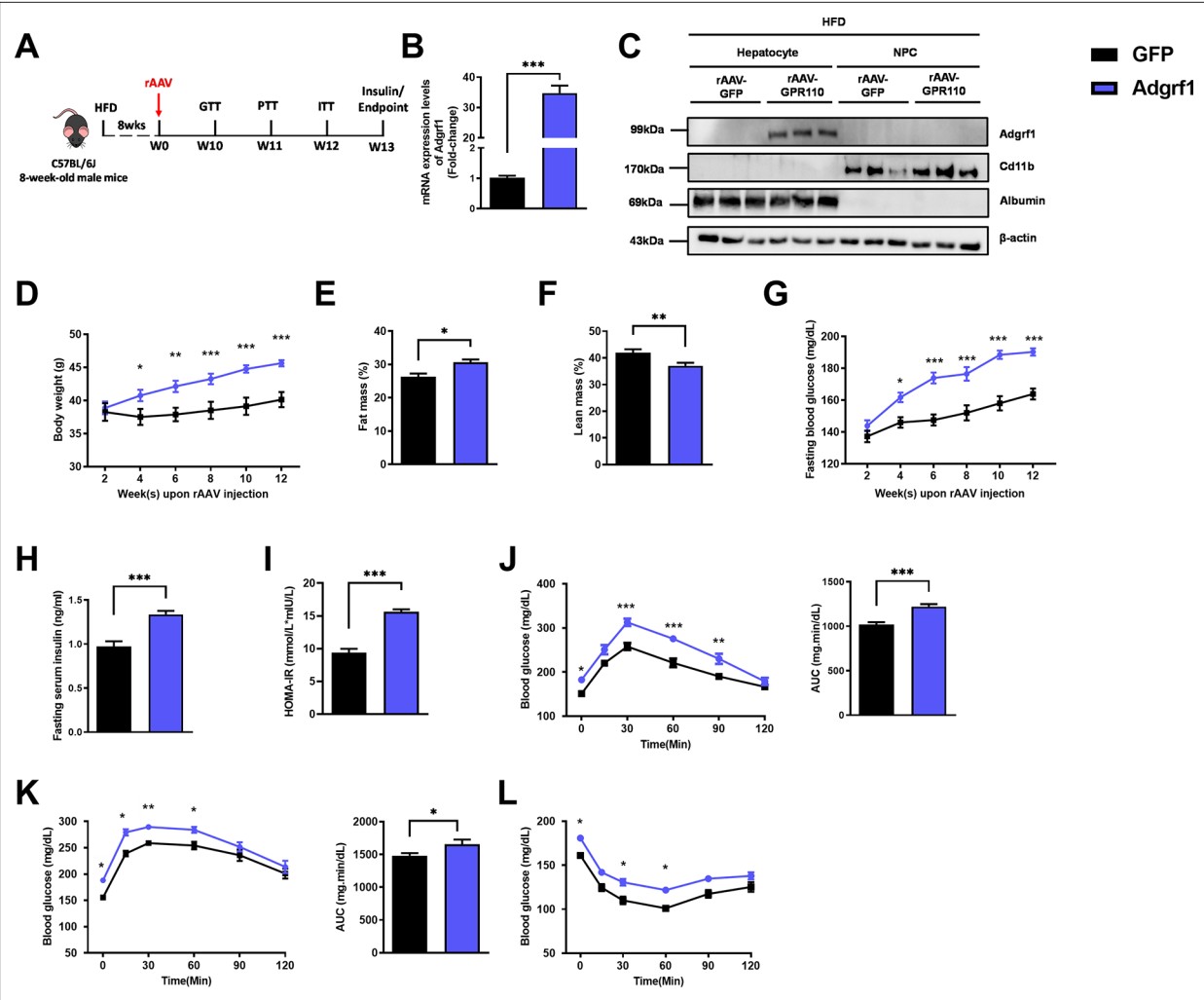

**Figure 2.** Overexpression of *Adgrf1* in hepatocytes exaggerates metabolic dysregulation by HFD treatment. Eight-week-old male C57BL/6J mice were infected with $3 \times 10^{11}$ copies of rAAV encoding *Adgrf1* (rAAV-*Adgrf1*, i.v.) or control (rAAV-*GFP*, i.v.) and received HFD feeding, respectively. (**A**) Schematic illustration of viral treatments. (**B**) Hepatic mRNA expression levels of *Adgrf1* from rAAV-*Adgrf1* mice liver in fractions of hepatocyte or NPCs isolated from HFD-fed mice livers as determined by RT-qPCR. (**C**) Immunoblotting analyses of *Adgrf1*, *Cd11b*, and albumin from rAAV-*Adgrf1* mice liver in factions of hepatocyte or NPC isolated from HFD-fed mice livers. Each lane is a sample from a different individual. (**D**) Body weight, (**E**) the percentage of fat mass, and (**F**) lean mass were assessed in different groups. (**G**) Fasting blood glucose levels were measured biweekly upon rAAV injection. (**H**) The fasting serum insulin level and (**I**) HOMA-IR index were measured and calculated according to the formula [Fasting blood glucose (mmol/l) × Fasting blood insulin (mIU/l)]/22.5 for the HFD-fed rAAV-*Adgrf1* or rAAV-*GFP* mice at the end of the experiment. (**J**) GTT (1 g/kg BW, left) and area under curve (AUC, right) of serum glucose at week 10. (**K**) PTT (1 g/kg BW, left) and AUC (right) of serum glucose at week 11. (**L**) ITT (0.5 U/kg BW, left) and AUC (right) of serum glucose at week 12. mRNA expression levels of the target genes were normalized to the expression of mouse *Gapdh*. rAAV-NC group was set as 1 for fold-change calculation. n = 8 per group. *Adgrf1*, G-protein-coupled receptor 110; STC, standard chow diet; HFD, high-fat diet; NPC, non-parenchymal cell; BW, body weight; GTT, glucose tolerance test; PTT, pyruvate tolerance test; ITT, insulin tolerance test; AUC, area under curve; NC, negative control; HOMA-IR, homeostasis model assessment-estimated insulin resistance. Data represents as mean ± SEM; repeated with three independent experiments; p-value analyzed by two-tailed Student's *t*-test. *p<0.05, **p<0.01, ***p<0.001.

The online version of this article includes the following source data and figure supplement(s) for figure 2:

**Source data 1.** Overexpression of Adgrf1 in hepatocytes exaggerates metabolic dysregulation by HFD treatment.

**Figure supplement 1.** Hepatic overexpression of *Adgrf1* in STC-fed mice exhibits mild metabolic abnormalities.

**Figure supplement 1—source data 1.** Hepatic overexpression of Adgrf1 in STC-fed mice exhibits mild metabolic abnormalities.

**Figure supplement 2.** Overexpression of Adgrf1 did not interfere adipose tissues and other metabolic phenotypes.

**Figure supplement 2—source data 1.** Overexpression of Adgrf1 did not interfere adipose tissues and other metabolic phenotypes.

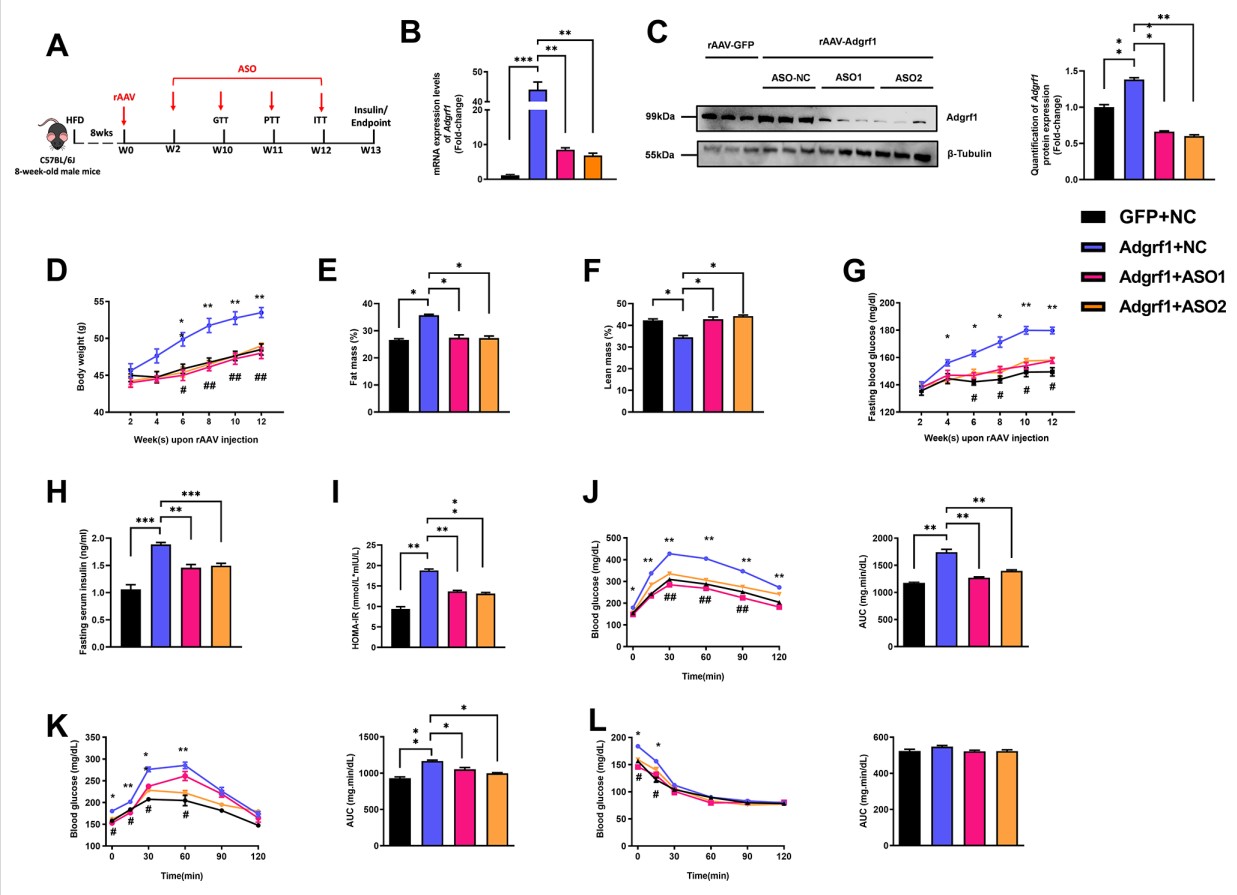

**Figure 3.** Deletion of hepatic *Adgrf1* protects against diet-induced glucose intolerance in *Adgrf1* overexpress mice. Eight-week-old male C57BL/6J mice were infected with either $3 \times 10^{11}$ copies of rAAV encoding *Adgrf1* (rAAV-*Adgrf1*, i.v.) or control (rAAV-*GFP*, i.v.) and two different sequences of *Adgrf1* ASO (ASO1-*Adgrf1*, ASO2-*Adgrf1*, 5 mg/kg, one dose per week, s.c.) or scrambled control (ASO-NC, s.c.) received HFD feeding, respectively. (**A**) Schematic illustration of viral treatments. (**B**) Hepatic mRNA expression levels of *Adgrf1* from different groups of mice received either *GFP*-NC, *Adgrf1*-NC, *Adgrf1*-ASO1, or *Adgrf1*-ASO2 fed with HFD, respectively, as determined by RT-qPCR analysis. (**C**) Left panel: immunoblotting analyses of Adgrf1 and β-tubulin from livers of HFD-fed rAAV-GFP or rAAV-Adgrf1 mice treated with either ASO-NC or ASO-Adgrf1. Each lane is a sample from a different individual. Right panel: quantification of protein expression levels of Adgrf1 and β-tubulin. Protein expression levels were normalized to the expression of β-tubulin. (**D**) BW was measured biweekly upon rAAV and ASO injection. (**E**) The percentage of fat mass and (**F**) the percentage of lean mass were measured at the end of the experiment. (**G**) The fasting blood glucose level of different groups was measured upon rAAV and ASO injection. (**H**) Fasting serum insulin level and (**I**) HOMA-IR index were measured and calculated according to the formula [Fasting blood glucose (mmol/l)×Fasting blood insulin (mIU/l)]/22.5 for the HFD-fed rAAV-*Adgrf1* or rAAV-*GFP* mice at the end of the experiment. (**J**) GTT (1 g/kg BW, left) and AUC (right) of serum glucose at week 10. (**K**) PTT (1 g/kg BW, left) and AUC (right) of serum glucose at week 11. (**L**) ITT (0.5 U/kg BW, left) and AUC (right) of serum glucose at week 12. mRNA expression levels of the target genes were normalized to the expression of mouse *Gapdh*. rAAV-NC group was set as 1 for fold-change calculation. n = 8 per group. STC, standard chow diet; HFD, high-fat diet; ASO, antisense oligonucleotides; BW, body weight; GTT, glucose tolerance test; PTT, pyruvate tolerance test; ITT, insulin tolerance test; AUC, area under curve; NC, negative control; HOMA-IR, homeostasis model assessment-estimated insulin resistance. Data represented as mean ± SEM; repeated with three independent experiments; p-value analyzed by two-tailed Student's *t*-test. *p<0.05, **p<0.01, ***p<0.001.

The online version of this article includes the following source data and figure supplement(s) for figure 3:

**Source data 1.** Deletion of hepatic Adgrf1 protects against diet-induced glucose intolerance in Adgrf1 overexpress mice.

**Figure supplement 1.** Hepatic knockdown of *Adgrf1* in STC-fed mice does not exhibit metabolic abnormalities.

**Figure supplement 1—source data 1.** Hepatic knockdown of Adgrf1 in STC-fed mice does not exhibit metabolic abnormalities.

(*Figure 3—figure supplement 1D*) and fasting glucose (*Figure 3—figure supplement 1E*), but slightly lowered insulin level (*Figure 3—figure supplement 1F*) and HOMR-IR (*Figure 3—figure supplement 1G*) compared to their littermates injected with the negative control – scrambled antisense oligonucleotides (ASO-NC). No difference in the changes of glucose levels in GTT (*Figure 3—figure*

*supplement 1H*), PTT (*Figure 3—figure supplement 1I*), and ITT (*Figure 3—figure supplement 1J*) for both ASO-*Adgrf1*s and ASO-NC groups under STC feeding conditions.

In contrast, chronic ASO-*Adgrf1* treatment for 4 wk significantly decreased their body weight (*Figure 3D*), fat mass ratio (*Figure 3E and F*), fasting glucose level (*Figure 3G*), fasting insulin level (*Figure 3H*), and HOMA-IR (*Figure 3F*) in HFD-fed rAAV-*Adgrf1* mice. In addition, treatment of ASO-*Adgrf1*s improved glucose tolerance, pyruvate tolerance, and insulin sensitivity in HFD-fed rAAV-*Adgrf1* mice as demonstrated by GTT (*Figure 3J*), PTT (*Figure 3K*), and ITT (*Figure 3L*) compared to ASO-NC controls. In consistent to overexpressing *Adgrf1* in livers, the depletion of hepatic *Adgrf1* by ASOs improves glucose homeostasis in HFD-fed mice.

## Treatment of ASO-*Adgrf1*s alleviates lipid abundance and liver damage in HFD-fed rAAV-*Adgrf1* mice

The circulating lipid profiles of the mice were also checked. HFD-fed rAAV-*Adgrf1* mice had higher circulating cholesterol (CHO; *Figure 4A*) and triglyceride (TG; *Figure 4B*) levels than HFD-fed rAAV-*GFP* littermates, but their circulating free fatty acid (FFA) levels were similar (*Figure 4C*). High-density lipoprotein (HDL) cholesterol level was decreased, and low-density lipoprotein (LDL) cholesterol level was increased in HFD-fed rAAV-*Adgrf1*-NC mice compared to chronic ASO-*Adgrf1* treatment group (*Figure 4D*).

Remarkably, chronic ASO-*Adgrf1* treatment could lower circulating levels of the liver enzymes aspartate transaminase (AST) and alanine aminotransaminase (ALT), as markers of liver damage and hepatoxicity, in HFD-fed rAAV-*Adgrf1*-NC mice (*Figure 4E*). After sacrificing the mice, the hepatic lipid profiles were examined. We found that the livers of HFD-fed rAAV-*Adgrf1* mice were significantly heavier (*Figure 4F*) and paler (*Figure 4G*, upper panels) than the livers of their rAAV-*GFP* littermates and ASO-*Adgrf1*-treated rAAV-GRP110 mice. Consistent with these observations, HFD-induced lipid accumulation within hepatocytes was substantially more abundant in the livers of HFD-fed rAAV-*Adgrf1* mice than the rAAV-*GFP* littermates as determined by hematoxylin and eosin (H&E) staining and Oil Red O staining (*Figure 4G*, upper and middle rows). Moreover, based on the Masson trichrome staining, more fiber extension and larger fibrous septa formation were observed for the liver samples of rAAV-*Adgrf1* mice compared to the livers from rAAV-*GFP* littermates (*Figure 4G*, lower row). These alterations were remarkably reduced after ASO-*Adgrf1* treatments (*Figure 4G*). Like the circulating lipid profiles mentioned above, treatment of ASO-*Adgrf1*s for 8 wk could improve the hepatic lipid profiles of rAAV-*Adgrf1* mice in terms of CHO (*Figure 4H*), TG (*Figure 4I*), and FFA (*Figure 4J*). Altogether, overexpression of hepatic *Adgrf1* in mice is sufficient to perturb lipid metabolism and hence the progression of NAFLD, especially in obese subjects.

## The metabolic dysregulation of rAAV-*Adgrf1* mice is correlated to the upregulation of *Scd1* expression

To reveal the molecular mechanism underlying the involvement of hepatic *Adgrf1* in NAFLD development, RNA-sequencing analysis was performed on RNA samples extracted from the livers of HFD-fed ASO-NC-treated rAAV-GPF, ASO-NC-treated rAAV-*Adgrf1*, and ASO-*Adgrf1*-treated rAAV-*Adgrf1* mice. In the search for the molecular processes for metabolisms, several lipid metabolism-related genes were altered (*Figure 5A and B*). We subsequently used RT-qPCR to confirm the RNA sequencing results (*Figure 5C*). Among them, we are particularly interested in stearoyl CoA desaturase 1 (*Scd1*), a key lipogenic enzyme responsible for the rate-limiting step in the synthesis of monounsaturated fatty acids (MUFAs), such as oleate and palmitoleate, by forming double bonds in saturated fatty acids (*Paton and Ntambi, 2009*). MUFAs serve as substrates for the synthesis of various kinds of lipids, and increases in *Scd1* activity are involved in the development of NAFLD, hypertriglyceridemia, atherosclerosis, and diabetes (*Kotronen et al., 2009*; *Mar-Heyming et al., 2008*). To confirm the role of Adgrf1 in lipogenesis, the de novo lipogenesis assay was performed. Hepatocytes isolated from the rAAV-Adgrf1 group showed the highest lipogenesis activity, while those from the control rAAV-GFP group had lower activity (*Figure 5D*). Furthermore, the lipogenic activity decreased in the hepatocytes isolated from rAAV-Adgrf1 mice treated with Adgrf1-specific ASOs compared to the control group (*Figure 5D*). In agreement with the RT-qPCR data mentioned above, the results revealed a direct correlation between the level of Adgrf1 expression and lipogenic activity.

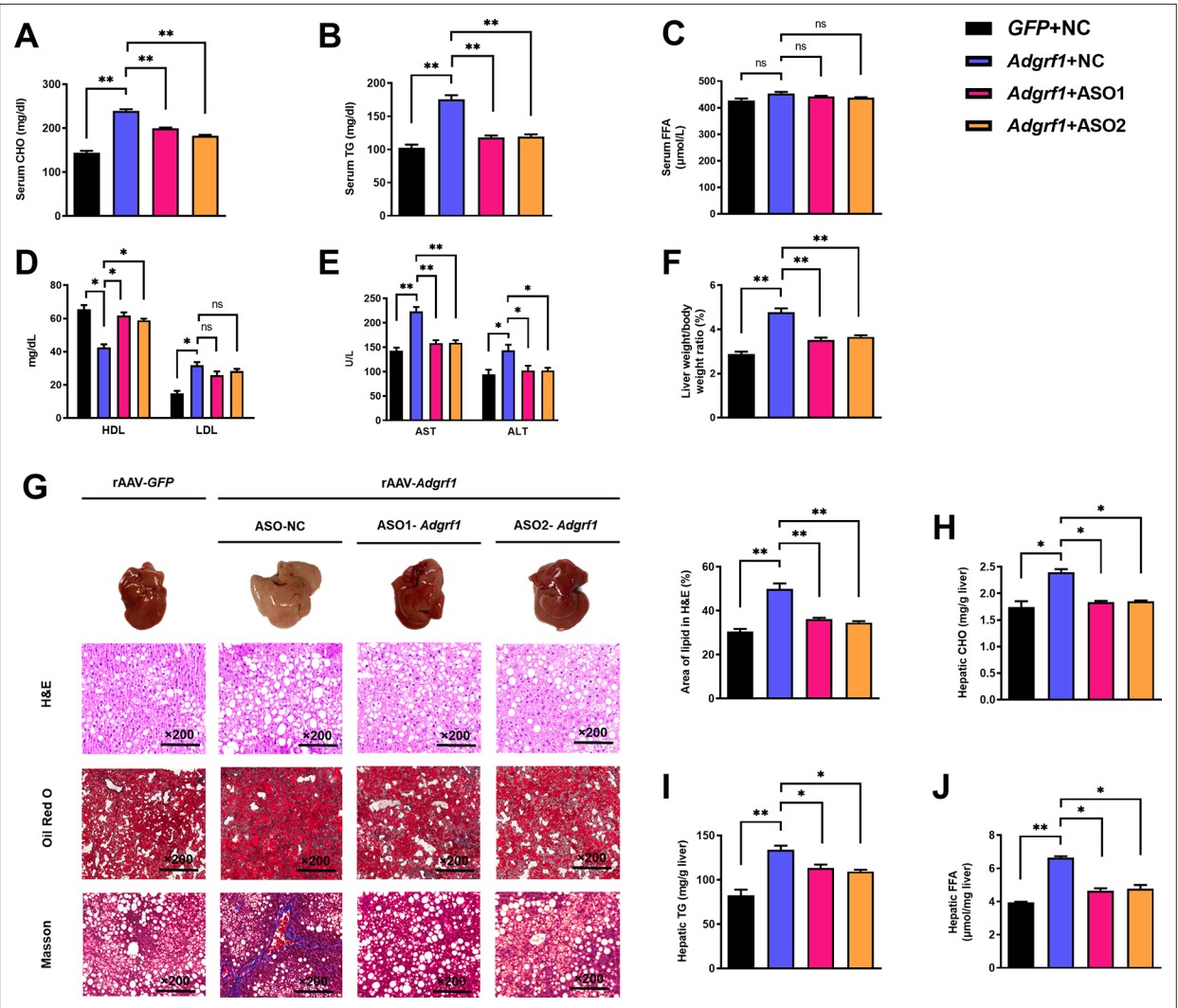

**Figure 4.** Upregulation of hepatic *Adgrf1* exaggerates liver steatosis in HFD-fed mice while downregulation of hepatic *Adgrf1* protects mice from diet-induced liver lipid accumulation. Eight-week-old male C57BL/6J mice were infected with either $3 \times 10^{11}$ copies of rAAV encoding *Adgrf1* (rAAV-*Adgrf1*, i.v.) or control (rAAV-NC, i.v.) and two different sequences of *Adgrf1* antisense oligonucleotides (ASO1-*Adgrf1*, ASO2-*Adgrf1*, 5 mg/kg, one dose per week, s.c.) or scrambled control (ASO-NC, s.c.) received HFD feeding, respectively. (**A**) Serum CHO, (**B**) serum TG, and (**C**) serum FFA levels were measured at week 13. (**D**) Serum HDL and LDL. (**E**) The levels of serum AST ALT. (**F**) The ratio of the liver weight against body weight was calculated after sacrificing the mice from four different groups. (**G**) Representative gross pictures of liver tissues (upper panels), representative images of H&E (middle panels) and Oil Red O (lower panels) staining of liver sections (200 μm). The percentage of lipid area according to H&E staining (right panel). (**H**) Hepatic CHO, (**I**) hepatic TG, and (**J**) hepatic FFA were normalized by the weight of liver samples used for lipid extraction. n = 8 per group. i.v., intravenous injection; s.c., subcutaneous injection; STC, standard chow diet; HFD, high-fat diet; ASO, antisense oligonucleotides; BW, body weight; CHO, cholesterol; TG, triglyceride; FFA, free fatty acid; HDL, high-density lipoprotein; LDL, low-density lipoprotein; AST, aspartate transaminase; ALT, alanine transaminase; H&E, hematoxylin-eosin. Data represented as mean ± SEM; repeated with three independent experiments; p-value analyzed by two-tailed Student's *t*-test. *p<0.05, **p<0.01, ***p<0.001.

The online version of this article includes the following source data for figure 4:

**Source data 1.** Upregulation of hepatic *Adgrf1* exaggerates liver steatosis in HFD-fed mice while downregulation of hepatic *Adgrf1* protects mice from diet-induced liver lipid accumulation.

## The upregulation of *Scd1* expression in liver is driven by the presence of *Adgrf1*

To confirm *Scd1* expression is induced by *Adgrf1*, in vitro assays were performed by using adenovirus-mediated *Adgrf1* expression system (ADV-*Adgrf1*) to overexpress *Adgrf1* in primary hepatocytes isolated from STC-fed mice. After infection, the expressions of *Scd1* mRNAs (*Figure 6A*) and protein

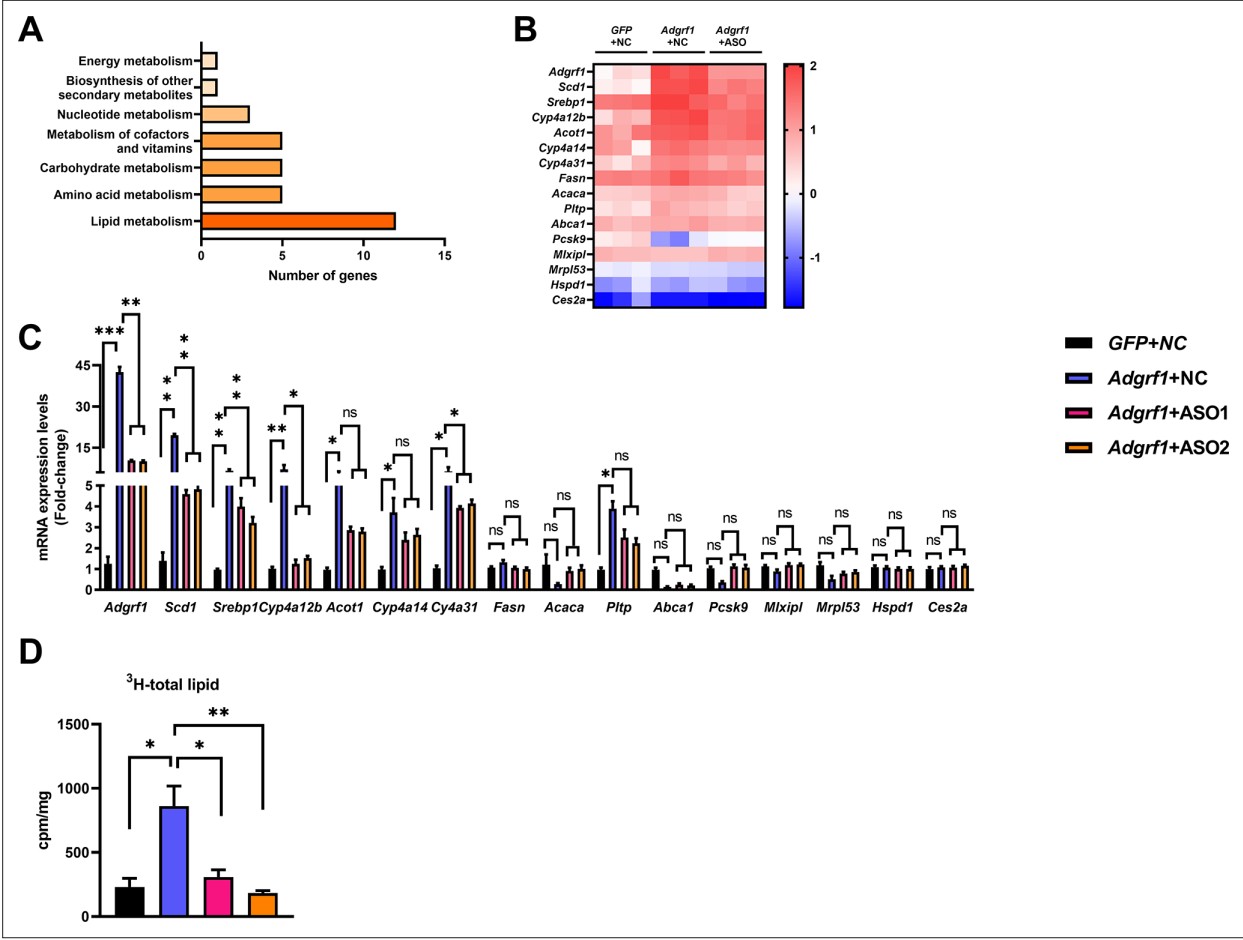

**Figure 5.** *Adgrf1* is a major regulator of hepatic lipid metabolism. Eight-week-old male C57BL/6J mice were infected with either $3 \times 10^{11}$ copies of rAAV encoding *Adgrf1* (rAAV-*Adgrf1*, i.v.) or control (rAAV-NC, i.v.) and two *Adgrf1* antisense oligonucleotides (ASO1-*Adgrf1*, ASO2-*Adgrf1*, 5 mg/kg, one dose per week, s.c.) or scrambled control (ASO-NC, s.c.) and received HFD feeding, respectively. Mice were sacrificed and mRNA of liver from each group were extracted and RNA-seq analysis was conducted. (**A**) KEGG pathway assay of differential mRNA transcripts in rAAV and ASO groups identified by RNA-seq. (**B**) Heat map shows the $\log_2$ scale fold change in the expression levels of a set of genes involved in lipid metabolism from RNA seq data of livers in HFD-fed mice treated by rAAV-*Adgrf1* or rAAV-*Adgrf1* plus *Adgrf1*-ASO1. n = 3 per group. (**C**) mRNA expression levels of genes according to the heat map from different groups of mice received either *GFP*-NC, *Adgrf1*-NC, *Adgrf1*-ASO1, or *Adgrf1*-ASO2 fed with HFD, respectively, as determined by RT-qPCR analysis, n = 6 mice per group. (**D**) De novo lipogenic activity was measured the ${}^3$H labeling of lipogenic Acetyl-CoA from 0.5 µCi ${}^3$H-acetate. ASO, antisense oligonucleotides. STC, standard chow diet; HFD, high-fat diet; i.v., intravenous injection; s.c., subcutaneous injection; ASO, antisense oligonucleotides; KEGG, Kyoto Encyclopedia of Genes and Genomes; GEO, gene expression omnibus; NAFLD, non-alcoholic fatty liver disease. Data represented as mean ± SEM; p-value analyzed by two-tailed Student's t-test. *p<0.05, **p<0.01, ***p<0.001.

The online version of this article includes the following source data for figure 5:

**Source data 1.** *Adgrf1* is a major regulator of hepatic lipid metabolism.

(*Figure 6B*) were dramatically induced, but not in control group, which was treated with ADV-*GFP*. In addition, the *Adgrf1* specific ASOs can not only knock down the *Adgrf1* expression, but also induced the *Scd1* expression by increasing *Adgrf1* in the ADV-*Adgrf1* primary hepatocytes (*Figure 6A and B*). In vitro luciferase reporter assay was performed to further validate the expression of *Scd1* is transcriptional regulated by *Adgrf1*. We constructed plasmid harboring luciferase gene driven by the mouse *Scd1* promoter (–2000 to +100) and transfected into HEK293T cells that were constructed. There was no change of luciferase activity of pGL3- *Scd1* promoter-luciferase-transfected HEK293T cells under the treatment of *Adgrf1* ligand DHEA, unless the HEK293T cells were pre-infected with adenovirus-overexpressing *Adgrf1* (ADV-*Adgrf1*; *Figure 6C*). The overexpression of *Adgrf1* in ADV-*Adgrf1*-infected cells and inductions of *Scd1* mRNA expression by treatment of DHEA were also validated by RT-qPCR (*Figure 6D*). The changes in hepatocyte lipid profiles by the expression level of *Adgrf1* and

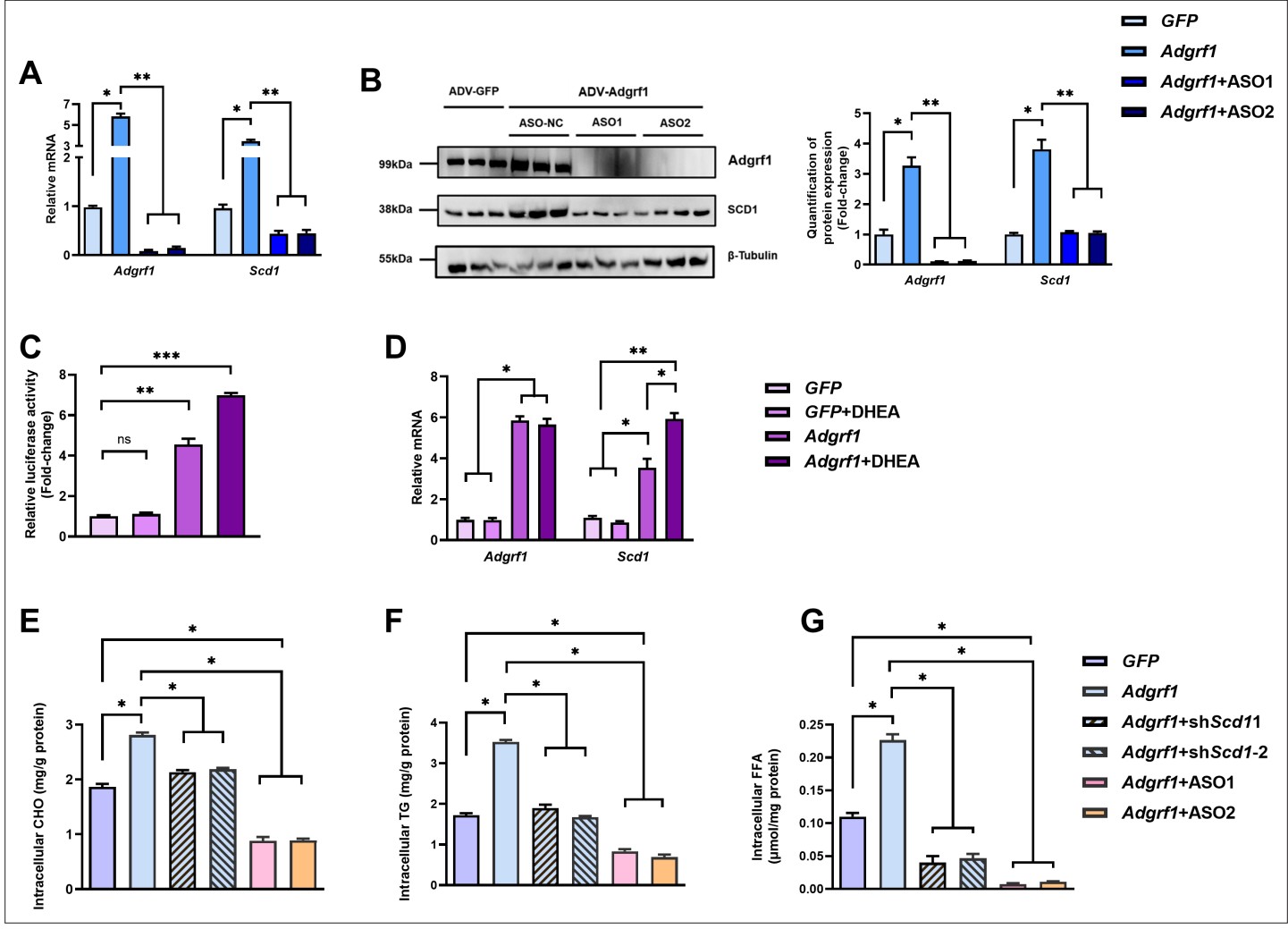

**Figure 6.** *Scd1* expression is regulated by *Adgrf1* in primary hepatocytes. Primary hepatocytes were isolated from 8-week-old male C57BL/6J mice with STC. (**A**) Primary hepatocytes were infected with either adenoviral vector expressing *Adgrf1* (ADV-*Adgrf1*) or control adenovirus expressing *GFP* (ADV-*GFP*) 24 hr after plating, followed by transfection with ASO1-*Adgrf1*, ASO2-*Adgrf1*, or ASO-NC for another 6 hr (n = 6). mRNA expression levels of *Adgrf1* and *Scd1* from different groups were assessed, as determined by RT-qPCR analysis. (**B**) Left panel: immunoblotting analysis for the expression level of Adgrf1 and Scd1 from different groups of primary hepatocytes. Right panel: quantification of protein expression levels of Adgrf1 and Scd1. Protein expression levels were normalized to the expression of β-tubulin. Each lane is a sample from a different plate. Right panel: quantification of protein expression levels of Adgrf1, Scd1, and β-tubulin. n = 3 per group. Protein expression levels were normalized to the expression of β-tubulin. The samples for GFP were set as 1 for fold-change calculation. (**C, D**) HEK293T cells were infected with pGL3-*Scd1* promoter-luciferase plasmid and adenoviral vector expressing *Adgrf1* (ADV-*Adgrf1*) or *GFP* (ADV-*GFP*) for 48 hr and DHEA was added to the transfected cells at the concentration of 100 μM for 24 hr. Cell lysates were used for (**C**) luciferase assay or (**D**) RT-qPCR analysis (n = 3). Lysates from the cell co-transfection with pGL3-*Scd1* promoter-luciferase plasmid and ADV-*GFP* without treatment of DHEA was set as 1 for fold-change calculation. (**E–G**) Primary hepatocytes were infected with either adenoviral vector expressing *Adgrf1* (ADV-*Adgrf1*) or control ADV-*GFP*, followed by transfecting with scramble or sh*Scd1*-1 or sh*Scd1*-2 plasmids for another 72 hr. Intracellular lipids were extracted and (**E**) CHO, (**F**) TG, and (**G**) FFA were assessed (n = 3). STC, standard chow diet; i.v., intravenous injection; s.c., subcutaneous injection; ASO, antisense oligonucleotides. CHO, cholesterol; TG, triglyceride; FFA, free fatty acid. Data represented as mean ± SEM; repeated with three independent experiments; p-value analyzed by two-tailed Student's *t*-test. *p<0.05, **p<0.01, ***p<0.001.

The online version of this article includes the following source data for figure 6:

**Source data 1.** *Scd1* expression is regulated by *Adgrf1* in primary hepatocytes.

*Scd1* were also checked. In agreement with the in vivo studies' findings, the overexpression of *Adgrf1* increased the intracellular CHO (*Figure 6E*), TG (*Figure 6F*), and FFA (*Figure 6G*). Their increases could be completely repressed by ASO against *Adgrf1* (*Figure 6E–G*) and partially repressed by overexpressing *Scd1*-specific shRNAs (*Figure 6F and G*). In summary, the transcription level of *Scd1* is regulated by *Adgrf1*. *Adgrf1* enhances the lipid accumulation by inducing *Scd1* expression.

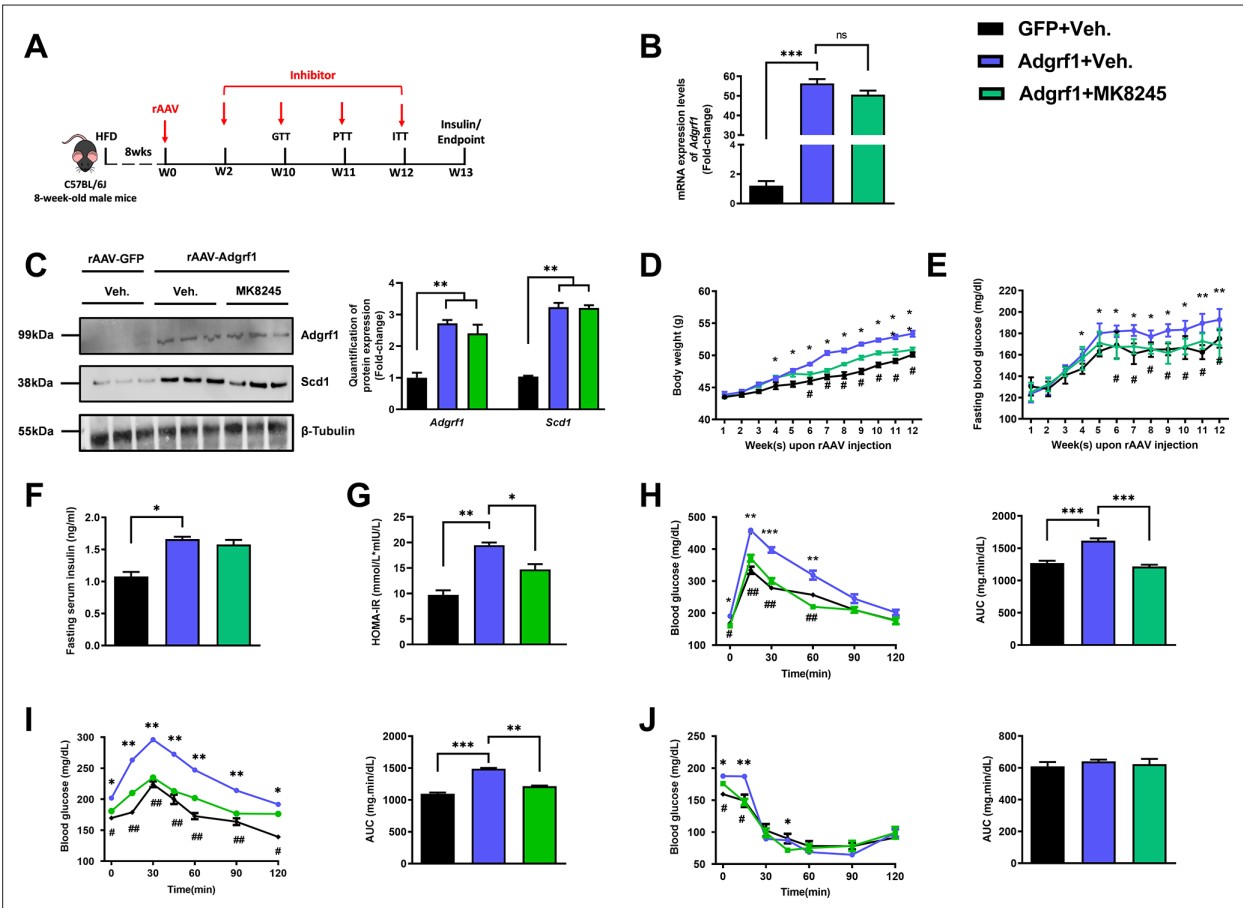

**Figure 7.** Inhibition of *Scd1* alleviates the glucose impairment in mice with hepatic *Adgrf1* overexpression. Eight-week-old male C57BL/6J mice were infected with either 3 × 10¹¹ copies of rAAV encoding *Adgrf1* (rAAV-*Adgrf1*, i.v.) or control (rAAV-*GFP*, i.v.) and *Scd1* inhibitor (MK8245, 10 mg/kg/week, p.o.) or inhibitor vehicle (inhibitor-Veh., p.o.) received HFD feeding. (**A**) Schematic illustration of viral treatments. (**B**) Hepatic mRNA expression levels of *Adgrf1* from different groups of mice received rAAV and inhibitor fed with HFD, respectively, as determined by RT-qPCR analysis. (**C**) Left panel: immunoblotting analysis for the hepatic protein expression level of Adgrf1 and Scd1 from different groups of mice fed with HFD. Right panel: quantification of protein expression levels of Adgrf1 and Scd1. Protein expression levels were normalized to the expression of β-tubulin. Each lane is a sample from a different individual. (**D**) BW and (**E**) fasting blood glucose level were measured at different weeks upon rAAV and inhibitor injection. (**F**) The fasting blood insulin level and (**G**) HOMA-IR index were measured and calculated according to the formula [Fasting blood glucose (mmol/l)×Fasting blood insulin (mIU/l)]/22.5 for the HFD-fed rAAV-*Adgrf1* or rAAV-*GFP* mice at the end of the experiment. (**H**) GTT (1 g/kg BW, left) and AUC (right) of serum glucose at the week of 10. (**I**) PTT (1 g/kg BW, left) and AUC (right) of serum glucose at week 11. (**J**) ITT (0.5 U/kg BW, left) and AUC (right) of serum glucose at week of 12. mRNA expression levels of the target genes were normalized to the expression of mouse *Gapdh*. rAAV-NC group was set as 1 for fold-change calculation. n = 8 per group. HFD, high-fat diet; i.v., intravenous injection; p.o., oral administration; ASO, antisense oligonucleotides; BW, body weight; GTT, glucose tolerance test; PTT, pyruvate tolerance test; ITT, insulin tolerance test; AUC, area under curve; NC, negative control; HOMA-IR, homeostasis model assessment-estimated insulin resistance. Data represented as mean ± SEM; repeated with three independent experiments; p-value analyzed by two-tailed Student's *t*-test. *p<0.05, **p<0.01, ***p<0.001.

The online version of this article includes the following source data for figure 7:

**Source data 1.** Inhibition of Scd1 alleviates the glucose impairment in mice with hepatic Adgrf1 overexpression.

## Inhibition of *Scd1* in rAAV-*Adgrf1* mice partially attenuates most metabolic dysregulations, especially lipid profiles

To examine whether the upregulation of hepatic *Scd1* was the cause of metabolic dysregulation in rAAV-*Adgrf1* mice, a liver-specific *Scd1* inhibitor MK8245 was used to alleviate the metabolic dysregulation by overexpressing *Adgrf1* in HFD-fed rAAV-*Adgrf1* mice (*Figure 7A*; *Iida et al., 2018*). Chronic treatment of MK8245 for 11 wk did not affect the expression of *Adgrf1* mRNA (*Figure 7B*) and protein (*Figure 7C*) levels in rAAV-GRP110 mice. In agreement with previous studies showing that the chronic treatment of this *Scd1* inhibitor improves various metabolic parameters including lipid and glucose profiles in various animal models (*Oballa et al., 2011*), treatment of MK8245 lowered the body weight (*Figure 7D*), improved glucose homeostasis in terms of fasting glucose level (*Figure 7E*) and HOMR-IR (*Figure 7G*), and performance in GTT (*Figure 7H*) and PTT (*Figure 7I*) of HFD-fed rAAV-*Adgrf1* mice compared to untreated littermates. But there was no change in insulin sensitivity as demonstrated by ITT (*Figure 7J*).

MK8245 treatment also lowered the circulating CHO (*Figure 8A*) and TG (*Figure 8B*) levels almost to the levels of HFD-fed rAAV-*GFP* mice, but there was no change in circulating FFA level (*Figure 8C*). A relatively higher HDL can be found in the MK8245 group, but there were no differences detected regarding the LDL level (*Figure 8D*). The AST and ALT levels were also alleviated in the MK8245 group compared to the rAAV-*Adgrf1* littermates (*Figure 8E*). MK8245 treatment partially reduced the liver weight (*Figure 8F*), degree of paleness, severity of fibrosis (*Figure 8G*), and lipid accumulations (*Figure 8H–J*). To conclude, treatment of MK8245 could improve the lipid profiles and alleviate metabolic dysregulation caused by overexpression of hepatic *Adgrf1* in mice.

## Expression of *Adgrf1* in liver is closely associated with hepatic steatosis in NAFLD patients

To evaluate the clinical relevance of the findings in mice, we first checked the expression level of *Adgrf1* in human liver from a publicized transcriptome dataset Gene Expression Omnibus (GEO; Profile # GDS4881) with human liver biopsy of different phases from control to NAFLD was checked (*Ahrens et al., 2013*). Healthy obese subjects without NAFLD had lower *Adgrf1* mRNA expression than healthy lean subjects, but obese NAFLD subjects had similar *Adgrf1* mRNA expression level as healthy lean subjects (*Figure 9A*). Subsequently, by using the same transcriptome dataset, the correlation in the expression level of *Adgrf1* and *Scd1* was investigated. The expression level of *Adgrf1* was positively correlated with *Scd1* in the liver ($r = 0.4635$, $p < 0.05$; *Figure 9B*).

To verify the observation, we performed immunohistochemistry staining with liver sections from biopsy-proven patients with mild, moderate, and severe NAFLD, respectively (*Supplementary file 4*). The degree of steatosis was determined by non-alcoholic steatohepatitis clinical research network (NASH CRN) scoring system (*Brunt et al., 1999*). Immunostaining analysis demonstrated that hepatic expression of *Adgrf1* protein was higher in the ones with severe steatosis than those with lower degree of NAFLD (*Figure 9C*). These data collectively suggest that *Adgrf1* expression level correlates to hepatic steatosis in humans as well.

To investigate the potential reason for higher hepatic Adgrf1 expression levels in NAFLD patients than in healthy obese individuals, we formulated the multiple-hit hypothesis, which suggests that liver inflammation may alter gene expression during NAFLD pathogenesis (*Buzzetti et al., 2016*). Supporting the hypothesis, it is found that the mRNA level of IL-1β, a key mediator of low-grade inflammation during NAFLD (*Mirea et al., 2018*), was significantly higher in NAFLD patients than in healthy obese individuals in the GEO (Profile # GDS4881) (*Figure 9D*). To validate this hypothesis, HFD-fed mice with either CCl4 or STZ to accelerate and exacerbate their NAFLD pathogenesis (*Zhang et al., 2020*; *Zhong et al., 2020*; *Hansen et al., 2017*) and measured the mRNA level of the hepatic inflammation marker, IL-1β. Remarkably, after treatment with either CCl4 or STZ, the expression of IL-1β mRNA in their livers increased four- to fivefold compared to the respective value of untreated HFD-fed mice (*Figure 9E*). Notably, the expression of hepatic Adgrf1 mRNA was also significantly increased after CCl4 or STZ treatment compared to the untreated HFD-fed mice (*Figure 9F*). Additionally, the Scd1 mRNA expression level in the CCl4 or STZ-treated HFD-fed mice was significantly higher than the respective value of untreated HFD-fed mice (*Figure 9G*). These results suggest that inflammation induced by either CCl4 or STZ treatment can increase the expression of hepatic Adgrf1, in addition to Scd1, which is known to be involved in hepatic lipid metabolism.

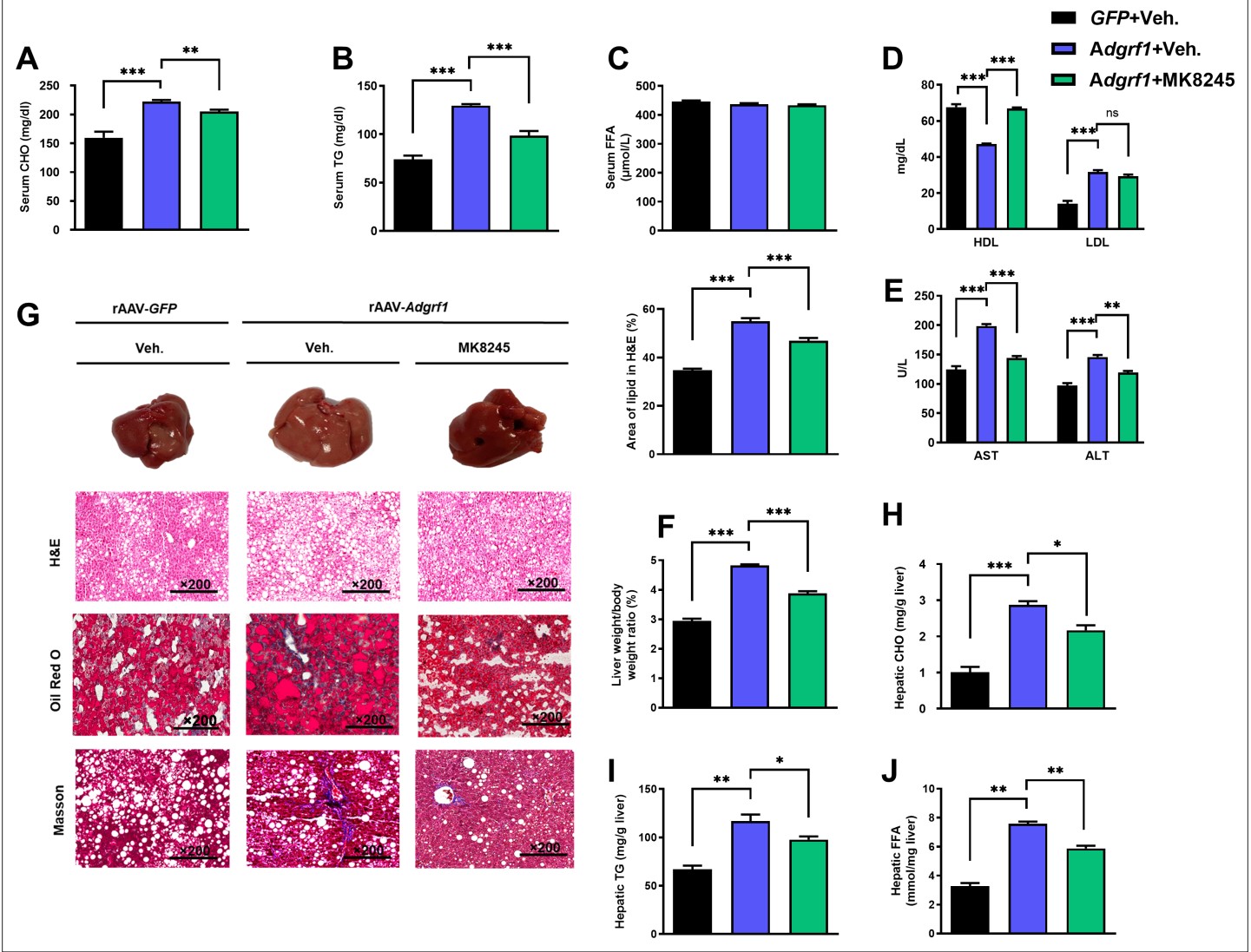

**Figure 8.** Inhibition of hepatic *Scd1* partially alleviates the severity of hepatic steatosis in *Adgrf1* overexpression mice. Eight-week-old male C57BL/6N mice were infected with either 3 × 10[11] copies of rAAV encoding *Adgrf1* (rAAV-*Adgrf1*, i.v.) or control (rAAV-*GFP*, i.v.) and administered with *Scd1* inhibitor (MK8245, 10 mg/kg, p.o.) or inhibitor vehicle (inhibitor-Veh., p.o.) received HFD feeding. (**A**) Serum CHO, (**B**) serum TG, and (**C**) serum FFA levels were measured at the end of experiment. (**D**) Serum HDL and LDL, (**E**) AST and ALT level of each group of mice were measured at the end of the experiment. (**F**) The ratio of the liver weight against body weight was calculated after sacrificing the mice from four different groups. (**G**) Representative gross pictures of liver tissues (upper panels), representative images of H&E (middle panels) and Oil Red O (lower panels) staining of liver sections (200 μm). The percentage of lipid area according to H&E staining (right panel). (**H**) Hepatic CHO, (**I**) hepatic TG, and (**J**) hepatic FFA were normalized by the weight of liver samples used for lipid extraction. n = 8 per group. STC, standard chow diet; HFD, high-fat diet; i.v., intravenous injection; p.o., oral administration; CHO, cholesterol; TG, triglyceride; FFA, free fatty acid; HDL, high-density lipoprotein; LDL, low-density lipoprotein; AST, aspartate transaminase; ALT, alanine transaminase; H&E, hematoxylin-eosin. Data represented as mean ± SEM; repeated with three independent experiments; p-value analyzed by two-tailed Student's *t*-test. *p<0.05, **p<0.01, ***p<0.001.

The online version of this article includes the following source data for figure 8:

**Source data 1.** Inhibition of hepatic Scd1 partially alleviates the severity of hepatic steatosis in Adgrf1 overexpression mice.

## Discussion

The present study has uncovered a previously unrecognized role of *Adgrf1* in regulating hepatic lipid metabolism. Firstly, we demonstrated that *Adgrf1* is highly expressed in liver of healthy subjects, and hepatic Adgrf1 is required for regulating lipid content in liver of diet-induced obese mice by both gain-of-function and loss-of-function approaches. Secondly, we found that the hepatic Adgrf1 expression level of healthy obese human and mouse are downregulated. The downregulation of hepatic

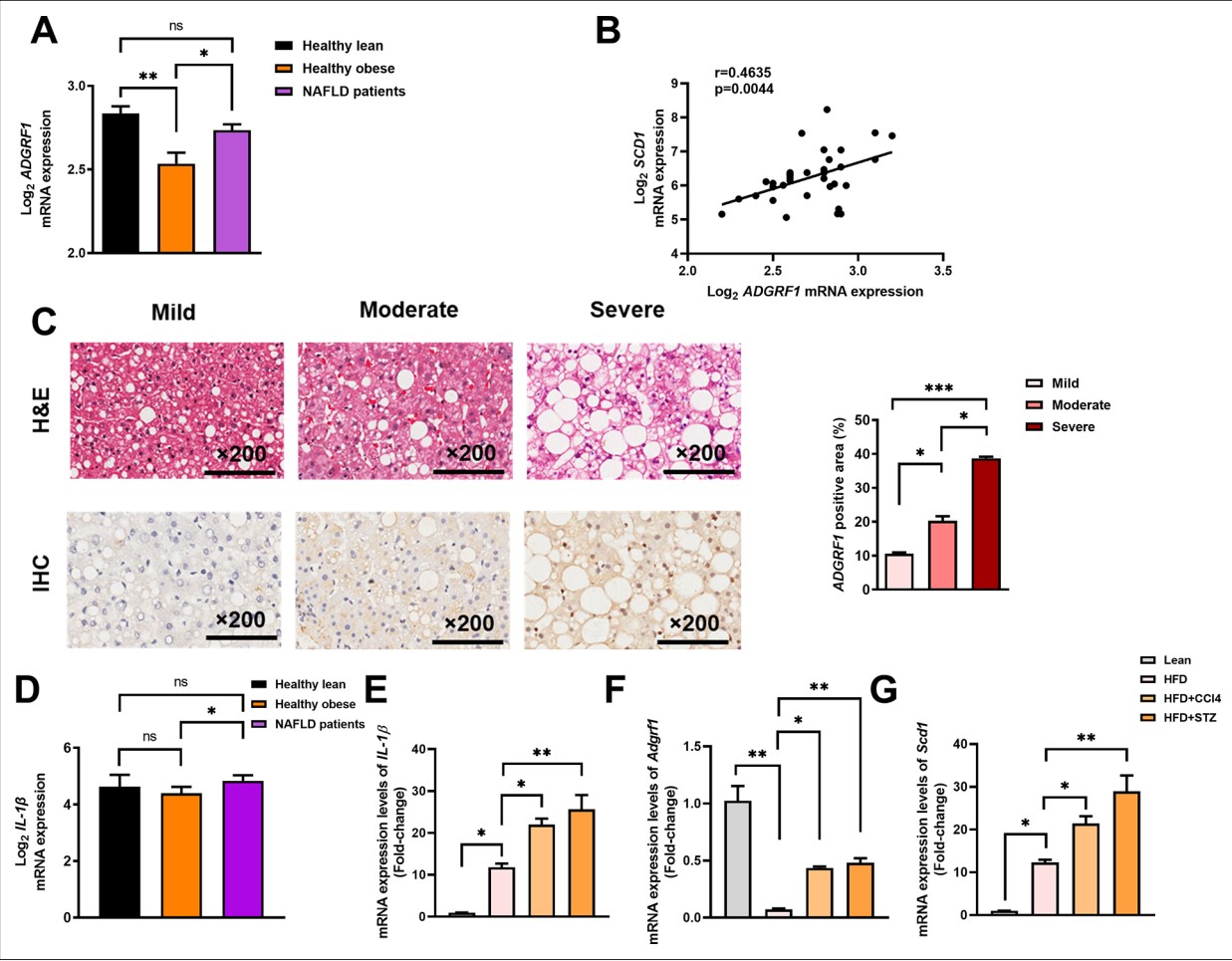

**Figure 9.** Hepatic expression of *ADGRF1* is upregulated in obese patients with hepatic steatosis when compared to those with normal liver morphology, which is positively associated with hepatic *SCD1* expression level. NAFLD patients have higher hepatic expression of *ADGRF1* accompanied with increased mRNA Scd1 expression. (**A**) Normalized Log$_2$ mRNA expression of *ADGRF1* in lean people without NAFLD (n = 12), obese people without NAFLD (n = 17), or obese patients with NAFLD (n = 8) according to the GEO database (GEO; Profile # GDS4881/8126820). (**B**) Correlation between *ADGRF1* and *SCD1* in liver of human subjects based on the GEO database. (**C**) Representative images of liver tissues with H&E staining (upper panels) and immunohistochemical staining (IHC) of *ADGRF1* (lower panels) from patients with different degree of NAFLD (200 μm). The percentage of *ADGRF1* positive area according to H&E staining (right panel). The percentage of *ADGRF1*-positive areas according to IHC staining (right panel); n = 3 per group. (**D**) Normalized Log2 mRNA expression of *IL-1β* in lean people without NAFLD (n = 12), obese people without NAFLD (n = 17), or obese patients with NAFLD (n = 8) according to the GEO database (GEO; Profile # GDS4881/8126820). (**E**) Hepatic mRNA expression levels of *IL-1β*, (**F**) *Adgrf1,* and (**G**) *Scd1* in either STC-fed mice or HFD-fed mice treated with CCl4 or STZ as determined by RT-qPCR. Data represented as mean ± SEM. p-Value analyzed by two-tailed Student's *t*-test. *p<0.05, **p<0.01, ***p<0.001.

The online version of this article includes the following source data for figure 9:

**Source data 1.** Hepatic expression of ADGRF1 is upregulated in obese patients with hepatic steatosis when compared to those with normal liver morphology, which is positively associated with hepatic SCD1 expression level.

*Adgrf1* expression level in obese subjects is a protective mechanism to prevent over-accumulation of lipid in liver. The argument mainly based on the findings that HFD-induced steatosis and liver injury were exacerbated in obese mice with high *Adgrf1* overexpression level, and knockdown hepatic *Adgrf1* alleviated the severity of obesity-induced NAFLD.

The pathophysiology of NAFLD is not well understood, but it is widely accepted that multiple factors, including inflammation, contribute to its acceleration and aggravation (*Buzzetti et al., 2016*). To explain the observation that hepatic Adgrf1 expression levels in NAFLD patients were higher than in healthy obese individuals, the expression levels of key inflammation markers in different mouse models were measured and it is found that IL-1β and Adgrf1 expression levels dramatically increased in subjects with more severe NAFLD conditions. It is interesting to note that the mechanism by which

hepatic Adgrf1 transcription is repressed in healthy obese individuals and induced in NAFLD patients remains to be explored. In addition, although obesity is commonly associated with NAFLD, up to 19% of lean Asians can also develop NAFLD (*VanWagner and Armstrong, 2018*). It would be interesting to investigate whether the expression levels of hepatic *Adgrf1* mRNA in these 'lean NAFLD' patients are higher than in lean healthy controls. This could provide further insights into the role of hepatic *Adgrf1* in NAFLD pathogenesis and help identify potential therapeutic targets for the treatment of NAFLD in lean individuals.

Subsequently, we performed RNA-sequencing analysis to investigate the mechanism of *Adgrf1*'s role in regulating lipogenesis and observed that the expression levels of *Scd1* mRNAs and protein were dramatically upregulated in the livers of rAAV-*Adgrf1* mice and repressed in *Adgrf1*-ASOs-treated rAAV-*Adgrf1* mice. Scd1 is the rate-limiting enzyme that catalyzes the conversion of saturated long-chain fatty acids into monounsaturated fatty acids, and its expression is tightly regulated by various parameters, including hormonal and nutrient factors (*Mauvoisin and Mounier, 2011*). A recent study demonstrated that Adgrf1 mediates palmitic acids to activate the mTor and Srebp1 pathways to promote fat synthesis in mammary gland tissues (*Zhang et al., 2023*). Here, Srebp1 is the key transcription factor for regulating Scd1 gene expression (*Zhu et al., 2019a*). Our RNA sequencing analysis also revealed that the expression of Srebp1 mRNA was dramatically upregulated in the livers of rAAV-Adgrf1 mice and repressed in Adgrf1-ASOs-treated rAAV-Adgrf1 mice, indicating that Adgrf1 might regulate Scd1 expression via Srebp1 in the liver. Importantly, the expression of hepatic Adgrf1 and Scd1 mRNAs can be further increased in the livers of subjects with more severe NAFLD, contributing to its acceleration and aggravation.

To confirm that the upregulation of hepatic *Scd1* expression levels was responsible for the changes in metabolic phenotype in the rAAV-*Adgrf1* mice, both in vivo and in vitro experiments were carried out using *Scd1* shRNAs and inhibitors. We found that pharmacologically inhibiting *Scd1* by MK8245 was sufficient to rescue the key metabolic dysregulations caused by *Adgrf1* overexpression in the liver. These results strongly support the conclusion that *Adgrf1* induces *Scd1* expression, leading to an increase in de novo lipogenesis in the liver and exacerbating obese-induced NAFLD.

Previous studies reported that *Scd1* global knockout (KO) mice showed improved insulin sensitivity, higher-energy metabolism, and more resistant to diet-induced obesity by the activation of lipid oxidation in addition to the reduction of triglyceride synthesis and storage (*Ntambi et al., 2002*; *Miyazaki et al., 2007*; *Miyazaki et al., 2009*). In addition, the *ob/ob* mice with *Scd1* mutations had significantly reduced storage of triglyceride and lower level of very low-density lipoprotein (VLDL) production (*Cohen et al., 2002*). Remarkably, liver-specific KO of *Scd1* was sufficient to reduce high-carbohydrate diet-induced adiposity with a significant reduction in hepatic lipogenesis and improved glucose tolerance (*Miyazaki et al., 2007*). Indeed, *Scd1* inhibition was proposed to be a therapeutic strategy for the treatment of metabolic syndrome (*Jiang et al., 2005*).

What are the potential advantages of targeting GRP110 rather than *Scd1* for the treatment of NAFLD? First of all, *Scd1* is highly expressed in various tissues, especially adipose tissues (*Ascenzi et al., 2021*). In addition, expression level and activity of *Scd1* are very tightly regulated (*Mauvoisin and Mounier, 2011*; *ALJohani et al., 2017*). Harmful consequences from inhibiting *Scd1* have been reported, such as the inhibition of fat mobilization in adipose tissues, and the promotion of proinflammatory and endoplasmic reticulum stress by accumulation *Scd1* substrates (*Brown and Rudel, 2010*; *Leung and Kim, 2013*; *Liu et al., 2011*; *Zou et al., 2020*). These findings clearly documented that optimal level of *Scd1* is required to maintain health. Secondly, in contrast to *Scd1*, according to the phenotypes of *Adgrf1* KO mice in previous studies (*Ma et al., 2017*; *Lee et al., 2016a*) and a dramatic reduction in *Adgrf1* in the livers of HFD-fed mice and health obese subjects, the hepatic *Adgrf1* may be dispensable in adults. Therefore, targeting hepatic *Adgrf1* is a potential safe treatment of NAFLD. Thirdly, according to the RNA-sequencing analysis, repressing *Adgrf1* can also regulate the expression of many other lipid metabolism genes. Unfortunately, *Adgrf1* antagonist is not available at this moment. As demonstrated, the ASO-based strategy is an alternative approach to knockdown the expression of *Adgrf1* in liver (*Dhuri et al., 2020*).

The current study focuses on *Adgrf1* in hepatic lipid metabolism. As mentioned above, *Adgrf1* is primarily understood to be an oncogene (*Lum et al., 2010*; *Liu et al., 2018*; *Zhu et al., 2019b*; *Shi and Zhang, 2017*; *Bhat et al., 2018*; *Ma et al., 2017*; *Nam et al., 2022*; *Abdulkareem et al., 2021*; *Sadras et al., 2017*; *Espinal-Enríquez et al., 2015*; *Harvey et al., 2010*). Notably, recent study

reported that deficiency of *Adgrf1* can decelerate carcinogen-induced hepatocarcinogenesis in adult mice (*Ma et al., 2017*). Coincidentally, high expression level of *Scd1* is also genetically susceptible to hepatocarcinogenesis (*Falvella et al., 2002*). It is highly possible that *Adgrf1* also accelerates carcinogenesis by inducing *Scd1* expression level. Therefore, a further experiment to check the *Scd1* expression level in the *Adgrf1* induced cancers may be conducted.

In summary, we present evidence demonstrating a novel role of hepatic *Adgrf1* in regulating lipid metabolism and explored the mechanism partially via regulation of *Scd1* expression level. As the amino acid sequencing of *Adgrf1* are highly conserved in humans and mice, targeting *Adgrf1* may serve as a novel therapeutic approach for the treatment of NAFLD patients.

## Acknowledgements

This work was supported by the National Natural Science Foundation of China (81870586), Area of Excellence (AoE/M-707/18), and General Research Fund (15101520) to CMW, and National Natural Science Foundation of China (82270941, 81974117) to SJ. We also thank Dr. Oscar Wong and Prof. Zou Xiang for critical review of this manuscript.

## Additional information

### Competing interests

Mengyao Wu, Chi-Ming Wong: has filed the following provisional patent: "Application of GPR110 as target for treating metabolic diseases"; US Provisional Filing Ref No. 63/370,948. The author has no other competing interests to declare. Steve Ting-Yuan Yeh: is affiliated with Ionis Pharmaceuticals, and holds stock at Ionis. The author has no other competing interests to declare. The other authors declare that no competing interests exist.

### Funding

| Funder | Grant reference number | Author |
|---|---|---|
| National Natural Science Foundation of China | 81870586 | Chi-Ming Wong |
| Hong Kong University Grants Committee, Area of Excellence | AoE/M-707/18 | Chi-Ming Wong |
| Hong Kong University Grants Committee, General Research Fund | 15101520 | Chi-Ming Wong |
| National Natural Science Foundation of China | 82270941 | Jia Sun |
| National Natural Science Foundation of China | 81974117 | Jia Sun |

The funders had no role in study design, data collection and interpretation, or the decision to submit the work for publication.

### Author contributions

Mengyao Wu, Data curation, Investigation, Methodology, Writing – original draft; Tak-Ho Lo, Investigation, Methodology, Writing – original draft; Liping Li, Resources, Validation; Jia Sun, Resources, Funding acquisition; Chujun Deng, Validation, Investigation; Ka-Ying Chan, Xiang Li, Validation, Methodology; Steve Ting-Yuan Yeh, Resources, Methodology, Writing – original draft; Jimmy Tsz Hang Lee, Conceptualization, Methodology; Pauline Po Yee Lui, Validation, Investigation, Writing – original draft; Aimin Xu, Conceptualization, Supervision, Writing – original draft; Chi-Ming Wong, Conceptualization, Data curation, Formal analysis, Supervision, Funding acquisition, Validation, Investigation, Methodology, Writing – original draft, Project administration

## Author ORCIDs
Mengyao Wu (iD) https://orcid.org/0000-0001-5748-5780
Pauline Po Yee Lui (iD) http://orcid.org/0000-0003-2446-6203
Chi-Ming Wong (iD) https://orcid.org/0000-0002-0025-7135

## Ethics
The human study is approved by the Zhujiang Hospital, Southern Medical University, Guangzhou, China (Number: 2019-KY-097-01).

All animal procedures were approved by the Animal Subjects Ethics Sub-Committee of the Hong Kong Polytechnic University and were conducted in accordance with the guidelines of the Centralized Animals Facilities with the ethics number 19-20/78-HTI-R-OTHERS.

## Decision letter and Author response
Decision letter https://doi.org/10.7554/eLife.85131.sa1
Author response https://doi.org/10.7554/eLife.85131.sa2

---

## Additional files

### Supplementary files
- MDAR checklist
- Supplementary file 1. List of primary antibodies.
- Supplementary file 2. List of primers used for RT-qPCR.
- Supplementary file 3. Expression of GPCRs in the liver of mice fed with either STC or HFD diet for 8 weeks by gene expression microarray analysis.
- Supplementary file 4. Expression of lipogenic genes in the liver of mice fed with either STC or HFD diet for 8 weeks by gene expression microarray analysis.
- Supplementary file 5. Raw data for baseline characteristics of study cohorts.

### Data availability
All data generated or analysed during this study are included in the manuscript and supplementary data are either in the mansucript or source data files.

The following previously published dataset was used:

| Author(s) | Year | Dataset title | Dataset URL | Database and Identifier |
|---|---|---|---|---|
| Ahrens M, Ammerpohl O, von Schönfels W, Kolarova J | 2013 | Postbariatric, morbidly obese patients with nonalcoholic fatty liver disease: liver biopsies | https://www.ncbi.nlm.nih.gov/sites/GDSbrowser?acc=GDS4881 | NCBI Gene Expression Omnibus, GDS4881 |

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
