## [Editor Report]

These valuable findings presented by Wu et al. advance our understanding in novel cell signaling regulators of hepatic metabolism. The evidence supporting these conclusions is solid, utilizing in vivo and in vitro gain- and loss-of-function studies. This work will be of interest to biologists working in the field of hepatic steatosis.

---

## [Decision Letter]

**Decision letter after peer review:**

Thank you for submitting your article "Amelioration of non-alcoholic fatty liver disease by targeting G protein-coupled receptor 110: A preclinical study" for consideration by *eLife*. Your article has been reviewed by 3 peer reviewers, one of whom is a member of our board of Reviewing Editors and the evaluation has been overseen by Mone Zaidi as the Senior Editor. The following individual involved in review of your submission has agreed to reveal their identity: Weiping Han (Reviewer #3).

Essential revisions:

1) Show the levels of hepatic lipids and lipogenic gene expression, including SCD-1, in liver tissues from NCD vs. HFD-fed mice.

2) Test whether GPR110 overexpression would affect adipose tissue. Along the same line, they have to carefully investigate the reason of increased body weight gain in GPR110 overexpressed mice (ex., food intake, and energy expenditure).

3) Measurement of de novo lipogenic activity using primary hepatocyte with GPR110 overexpression or knockdown would be valuable to affirm the authors' proposed model.

4) Discuss potential the molecular mechanisms how GPR110 signaling could enhance SCD-1 transcription.

*Reviewer #1 (Recommendations for the authors):*

This is a well written and rigorously conducted project. My only recommendation for the author is to expand on the disconnect between murine diet-induced obesity and human steatosis with regards to expression of GPR110. In the mouse model, obesity impairs the expression of GPR110, whereas human steatosis enhances GPR110 expression. The authors should elaborate more thoroughly on this disconnect. Specifically, is downregulation of GPR110 in mice compensatory? Is this mechanism lost in humans?

*Reviewer #2 (Recommendations for the authors):*

Although it is of interest that GPR110/SCD-1 axis could modulate hepatic lipid levels, the underlying mechanisms are not clear. To support the authors' claim, it is necessary to validate the direct role of GPR110 in hepatic lipid metabolism and the detailed mechanisms of which GPR110 signaling could modulate SCD-1 transcription.

Comments/typos

1. Please provide detailed information for GalNAc-siGPR110 design (siRNA sequence, location of GalNAc, and RNA modification).

2. In page 2, knockdown GPR110 can reduced the cell migration → knockdown of GPR110 can reduce the cell migration.

3. In page 2, Reveal a new mechanism regulation the progression of NAFLD → Reveal a new mechanism regulating the progression of NAFLD.

4. Figure 5C: Cy4a12b -> Cyp4a12b.

*Reviewer #3 (Recommendations for the authors):*

The authors may examine whether GPR110 plays a more critical role at the early stage of NAFLD development, while possibly a second factor is more important for the late stage of NAFLD.

---

## [Author Response]

Essential revisions:1) Show the levels of hepatic lipids and lipogenic gene expression, including SCD-1, in liver tissues from NCD vs. HFD-fed mice.

Thank you for your comment. The levels of key hepatic lipids and lipogenic gene expression in liver tissues from NCD and HFD-fed mice in the supplementary Table 4 on page 62 to show the impact on hepatic lipid metabolism by altering gene expression and promoting lipid accumulation in the liver.

2) Test whether GPR110 overexpression would affect adipose tissue. Along the same line, they have to carefully investigate the reason of increased body weight gain in GPR110 overexpressed mice (ex., food intake, and energy expenditure).

Thank you for your comment. To investigate the effect of our liver-specific GPR110 overexpression system on adipose GPR110 expression, RT-qPCRs were performed. The results showed that GPR110 was overexpressed in the hepatocytes of livers (Figure 2B-C and S1C), but not in adipose tissues (Supplementary Figure 3A on page 50 in the revised manuscript) in the rAAV-GPR110 mice compared to the control group. In addition, the Ct value of adipose GPR110 mRNAs were over 40. It shall be due to the liver-specific thyroxine binding globulin (TBG) promoter was used to drive the expression of GPR110 in the liver via our rAAV-mediated gene expression system.

Metabolic phenotyping using a metabolic cage system revealed no significant differences in daily pedestrian locomotion, distance in cage locomotion, energy expenditure, food intake, water intake, or the respiratory exchange ratio between the rAAV-GFP and rAAV-GPR110 groups (Figure S3C-H on page 50), indicating that these factors did not contribute to the changes in fat mass.

We believe that the increased body weight gain in HFD-fed rAAV-GPR110 mice is mainly due to dysregulation of hepatic lipid metabolism and hence subsequent accumulation of fat mass. Similar phenotyping has been observed in mice with dysregulation of key hepatic genes involved in lipid metabolism (e.g., AMPK, PGC1a, SREBP-1c, Acat2, CD36, PPARα … etc.) [1-9]. Please refer to the following papers for further information.

1. Sachithanandan, N., et al., Liver‐specific suppressor of cytokine signaling‐3 deletion in mice enhances hepatic insulin sensitivity and lipogenesis resulting in fatty liver and obesity 1. Hepatology, 2010. 52(5): p. 1632-1642.

2. Morris, E.M., et al., Reduced liver-specific PGC1a increases susceptibility for short-term diet-induced weight gain in male mice. Nutrients, 2021. 13(8): p. 2596.

3. Ma, Z., et al., Hepatic Acat2 overexpression promotes systemic cholesterol metabolism and adipose lipid metabolism in mice. Diabetologia, 2023. 66(2): p. 390-405.

4. Wilson, C.G., et al., Hepatocyte-specific disruption of CD36 attenuates fatty liver and improves insulin sensitivity in HFD-fed mice. Endocrinology, 2016. 157(2): p. 570-585.

5. Mao, J., et al., Liver-specific deletion of acetyl-CoA carboxylase 1 reduces hepatic

triglyceride accumulation without affecting glucose homeostasis. Proceedings of the National Academy of Sciences, 2006. 103(22): p. 8552-8557.

6. Chakravarthy, M.V., et al., “New” hepatic fat activates PPARα to maintain glucose, lipid, and cholesterol homeostasis. Cell metabolism, 2005. 1(5): p. 309-322.

7. Dirkx, R., et al., Absence of peroxisomes in mouse hepatocytes causes mitochondrial and ER abnormalities. Hepatology, 2005. 41(4): p. 868-878.

8. Knebel, B., et al., Liver-specific expression of transcriptionally active SREBP-1c is associated with fatty liver and increased visceral fat mass. PLoS One, 2012. 7(2): p. e31812.

9. Garcia, D., et al., Genetic Liver-Specific AMPK Activation Protects against Diet-Induced Obesity and NAFLD. Cell Rep, 2019. 26(1): p. 192-208 e6.

3) Measurement of de novo lipogenic activity using primary hepatocyte with GPR110 overexpression or knockdown would be valuable to affirm the authors' proposed model.

Thank you for your comment. To investigate the effect of GPR110 expression level on de novo lipogenic activity in primary hepatocytes, we used stable isotopes ^3^H-acetate to measure their de novo lipogenic activities. The data are presented in Figure 5D on page 36 of the revised manuscript. The results indicate a direct correlation between GPR110 expression level and de novo lipogenic activity. The highest ^3^H-total lipid activity was detected in hepatocytes isolated from the rAAV-GPR110 group compared to the control rAAV-GFP group, while the activity was decreased in hepatocytes isolated from rAAV-GPR110 mice treated with GPR110-specific ASOs compared to the control group. These findings suggest that the expression level of GPR110 plays a crucial role in de novo lipogenic activity.

4) Discuss potential the molecular mechanisms how GPR110 signaling could enhance SCD-1 transcription.

Thank you for your comment. A study published on March 22, 2022, has shown that GPR110 mediates palmitic acid to activate the mTOR and SREBP1 pathways, promoting milk protein and fat synthesis in mammary gland tissues [10]. Our RNA sequencing results revealed a correlation between the expression of hepatic SREBP1 and GPR110. Therefore, we added the mRNA levels of SREBP1 in our experiments, presented in Figure 5B-C on page 36 of the revised manuscript. The expression level of SREBP1 was found to be higher in the GPR110 overexpression group and decreased after using ASOs to knockdown hepatic GPR110 levels. Additionally, we discussed this potential the molecular mechanism of GPR110 in regulating hepatic lipid metabolism through the SREBP1-SCD1 pathway in the revised version of the manuscript on page 21 line 455-464.

Reviewer #1 (Recommendations for the authors):This is a well written and rigorously conducted project. My only recommendation for the author is to expand on the disconnect between murine diet-induced obesity and human steatosis with regards to expression of GPR110. In the mouse model, obesity impairs the expression of GPR110, whereas human steatosis enhances GPR110 expression. The authors should elaborate more thoroughly on this disconnect. Specifically, is downregulation of GPR110 in mice compensatory? Is this mechanism lost in humans?

Thank you for your positive feedback and for highlighting areas where we can improve. Our study hypothesized that downregulating GPR110 is a protective mechanism against fatty liver development in both mouse and human. Similar to our findings in mice, we observed downregulation of hepatic GPR110 in healthy obese humans compared to lean humans, and a significant increase or restoration of hepatic GPR110 expression levels in NAFLD patients compared to both healthy obese humans and lean humans (see Figure 9A). However, the degree of change in GPR110 expression levels in human livers was not as pronounced as in mice (see Figure 1E-F). This could be due to several potential reasons, such as: (1) it is difficult to find humans with a BMI equivalent to mice treated with more than 12 weeks of HFD [11-15]; (2) we used RT-qPCR to determine hepatic GPR110 expression in mice, while microarrays were used to determine hepatic GPR110 expression in humans in previous study; and (3) furthermore, it's worth noting that the y-axis scales for these methods are different, and the human data from the microarray analysis is presented on a log2 scale, which may affect the impression of the changes of GPR110 mRNA expression in human appear to be less obvious.

Furthermore, inspired by the comments from reviewers, we hypothesized that pathological factors such as liver damage and inflammation in the livers of NAFLD patients may induce hepatic GPR110 expression. We also observed that the key inflammation marker, IL-1β, which mediates low-grade inflammation during NAFLD, was increased in our human study [5]. To mimic these pathological conditions, we induced liver damage and inflammation in the livers of HFD-fed mice using either CCl4 or streptozotocin (STZ) [16-20]. As shown in Figure 9 on page 44 of the revised manuscript, treatment with either CCl4 or STZ significantly increased the expression levels of the key hepatic inflammation marker IL-1β, GPR110, and SCD1 in the livers of the CCl4- or STZ-treated HFD-fed mice compared to their controls without any CCl4 or STZ treatment (Figure 9D-G). In summary, we believe that the mechanism is conserved between humans and mice, whereby the expression of hepatic GPR110 should be repressed in healthy obese individuals to prevent or delay the development of fatty liver. Pathological factors such as inflammation and liver damage may enhance GPR110 expression in NAFLD patients, thereby accelerating lipid accumulation in their livers through the SREBP1/SCD1 pathway.

Reviewer #2 (Recommendations for the authors):Although it is of interest that GPR110/SCD-1 axis could modulate hepatic lipid levels, the underlying mechanisms are not clear. To support the authors' claim, it is necessary to validate the direct role of GPR110 in hepatic lipid metabolism and the detailed mechanisms of which GPR110 signaling could modulate SCD-1 transcription.

Thank you for the comment. As mentioned in the response to your major comment #4, in line with the recent publication demonstrated that GPR110 mediates palmitic acids to activate the mTOR and SREBP1 pathways to promote milk protein and fat synthesis in mammary gland tissues, we did observe that the expression of key SCD1 transcription factor SREBP1 were correlated to the expression of GPR110. In brief, the expression level of SREBP1 level was found in the GPR110 overexpression group, and the level decreased after using ASOs to knockdown hepatic GPR110 levels. These findings suggest that GPR110 shall regulate hepatic lipid metabolism through the SREBP1-SCD1 pathway, and we added the mRNA levels of SREBP1 in my experiments in Figure 5B-C on page 36 of the revised manuscript.

Comments/typos1. Please provide detailed information for GalNAc-siGPR110 design (siRNA sequence, location of GalNAc, and RNA modification).

Please find the sequences and modification information of the ASOs in Author response table 1.

**Author response table 1. sa2table1:** 

	Sequence	GalNac position	Position of 2’ sugar modification
ASO1	AGGAAAATTTCGCTGA	5’-end	3*-10-3*
ASO2	GAATTTTAGGACTTGC	5’-end	3*-10-3*

2. In page 2, knockdown GPR110 can reduced the cell migration → knockdown of GPR110 can reduce the cell migration.

Thank you for pointing out the mistake. The letter 'd' was missing in the word 'reduced'. The correct sentence should be 'knockdown of GPR110 can reduce the cell migration'. Please refer to page 3, lines 31-32 for further clarification.

3. In page 2, Reveal a new mechanism regulation the progression of NAFLD → Reveal a new mechanism regulating the progression of NAFLD.

Thank you for pointing out the mistake. The word “regulation” was replaced by “regulating”. Please refer to page 3 line 50 for further clarification.

4. Figure 5C: Cy4a12b -> Cyp4a12b.

Thank you for pointing out the mistake. The letter “p” was added. Please refer to Figure 5B on page 36.

Reviewer #3 (Recommendations for the authors):The authors may examine whether GPR110 plays a more critical role at the early stage of NAFLD development, while possibly a second factor is more important for the late stage of NAFLD.

Thank you for the comment. We believe GPR110 plays a more critical role in the early stage of NAFLD, especially for the NAFLD patients beginning with inflammation or other insults that induce the NAFLD pathogenesis. Our argument mainly based on GPR110 is relative are relatively lower in the health obese subject than NAFLD patients, and one of the key differences these two groups is key inflammation marker Il-1β was higher detected in NAFLD patients as compared to health obese subjects. In addition, overexpression of GPR110 induced the expression of hepatic lipids and lipogenic genes and knockdown GPR110 in the rAAV-GPR110 mice can ameliorate of the progression of their NAFLD. The discussion of the important of GPR110 in NAFLD was presented on 20, lines 434-446.

References

1. Sachithanandan, N., et al., Liver‐specific suppressor of cytokine signaling‐3 deletion in mice enhances hepatic insulin sensitivity and lipogenesis resulting in fatty liver and obesity 1. Hepatology, 2010. 52(5): p. 1632-1642.

2. Morris, E.M., et al., Reduced liver-specific PGC1a increases susceptibility for short-term diet-induced weight gain in male mice. Nutrients, 2021. 13(8): p. 2596.

3. Ma, Z., et al., Hepatic Acat2 overexpression promotes systemic cholesterol metabolism and adipose lipid metabolism in mice. Diabetologia, 2023. 66(2): p. 390-405.

4. Wilson, C.G., et al., Hepatocyte-specific disruption of CD36 attenuates fatty liver and improves insulin sensitivity in HFD-fed mice. Endocrinology, 2016. 157(2): p. 570-585.

5. Mao, J., et al., Liver-specific deletion of acetyl-CoA carboxylase 1 reduces hepatic triglyceride accumulation without affecting glucose homeostasis. Proceedings of the National Academy of Sciences, 2006. 103(22): p. 8552-8557.

6. Chakravarthy, M.V., et al., “New” hepatic fat activates PPARα to maintain glucose, lipid, and cholesterol homeostasis. Cell metabolism, 2005. 1(5): p. 309-322.

7. Dirkx, R., et al., Absence of peroxisomes in mouse hepatocytes causes mitochondrial and ER abnormalities. Hepatology, 2005. 41(4): p. 868-878.

8. Knebel, B., et al., Liver-specific expression of transcriptionally active SREBP-1c is associated with fatty liver and increased visceral fat mass. PLoS One, 2012. 7(2): p. e31812.

9. Garcia, D., et al., Genetic Liver-Specific AMPK Activation Protects against Diet-Induced Obesity and NAFLD. Cell Rep, 2019. 26(1): p. 192-208 e6.

10. Zhang, M., et al., Comparative Transcriptomic Analysis of Mammary Gland Tissues Reveals the Critical Role of GPR110 in Palmitic Acid-Stimulated Milk Protein and Fat Synthesis. British Journal of Nutrition, 2023: p. 1-32.

11. Yazdi, F.T., S.M. Clee, and D. Meyre, Obesity genetics in mouse and human: back and forth, and back again. PeerJ, 2015. 3: p. e856.

12. Clee, S.M. and A.D. Attie, The genetic landscape of type 2 diabetes in mice. Endocrine reviews, 2007. 28(1): p. 48-83.

13. Gordon-Larsen, P., et al., Synergizing Mouse and Human Studies to Understand the Heterogeneity of Obesity. Adv Nutr, 2021. 12(5): p. 2023-2034.

14. Speakman, J., et al., The contribution of animal models to the study of obesity. Laboratory animals, 2008. 42(4): p. 413-432.

15. Kebede, M.A. and A.D. Attie, Insights into obesity and diabetes at the intersection of mouse and human genetics. Trends in Endocrinology and Metabolism, 2014. 25(10): p. 493-501.

16. Zhang, G., et al., Carbon tetrachloride (CCl4) accelerated development of non-alcoholic fatty liver disease (NAFLD)/steatohepatitis (NASH) in MS-NASH mice fed western diet supplemented with fructose (WDF). BMC gastroenterology, 2020. 20(1): p. 1-13.

17. Zhu, N., et al., Metabolomic study of high-fat diet-induced obese (DIO) and DIO plus CCl4induced NASH mice and the effect of obeticholic acid. Metabolites, 2021. 11(6): p. 374.

18. Hansen, H.H., et al., Mouse models of nonalcoholic steatohepatitis in preclinical drug development. Drug discovery today, 2017. 22(11): p. 1707-1718.

19. Tsuneyama, K., et al., Animal models for analyzing metabolic syndrome‐associated liver diseases. Pathology international, 2017. 67(11): p. 539-546.

20. Afrin, R., et al., Curcumin ameliorates liver damage and progression of NASH in NASH-HCC mouse model possibly by modulating HMGB1-NF-κB translocation. International immunopharmacology, 2017. 44: p. 174-182.

21. Zhu, X., et al., Berberine attenuates nonalcoholic hepatic steatosis through the AMPK- SREBP-1c-SCD1 pathway. Free Radical Biology and Medicine, 2019. 141: p. 192-204.

22. Ma, B., et al., Gpr110 deficiency decelerates carcinogen-induced hepatocarcinogenesis via activation of the IL-6/STAT3 pathway. Am J Cancer Res, 2017. 7(3): p. 433-447.

23. Mauvoisin, D. and C. Mounier, Hormonal and nutritional regulation of SCD1 gene expression. Biochimie, 2011. 93(1): p. 78-86.

24. Lounis, M.A., et al., Oleate activates SREBP-1 signaling activity in SCD1-deficient hepatocytes. American Journal of Physiology-Endocrinology and Metabolism, 2017. 313(6): p. E710-E720.